# Reconciling Privacy and Explainability in High-Stakes: A Systematic Inquiry

**Supriya Manna**                                                    *reachsmanna@gmail.com*
*SRM University AP, India*

**Niladri Sett**                                                     *settniladri@gmail.com*
*SRM University AP, India*

**Reviewed on OpenReview:** *https://openreview.net/forum?id=DQqdjPcE6g*

## Abstract

Deep learning's preponderance across scientific domains has reshaped high-stakes decision-making, making it essential to follow rigorous operational frameworks that include both Right-to-Privacy (RTP) and Right-to-Explanation (RTE). This paper examines the complexities of combining these two requirements. For RTP, we focus on 'Differential privacy' (DP), which is considered the current gold standard for privacy-preserving machine learning due to its strong quantitative guarantee of privacy. For RTE, we focus on post-hoc explainers: they are the go-to option for model auditing as they operate independently of model training. We formally investigate DP models and various commonly-used post-hoc explainers: how to evaluate these explainers subject to RTP, and analyze the intrinsic interactions between DP models and these explainers. Furthermore, our work throws light on how RTP and RTE can be effectively combined in high-stakes applications. Our study concludes by outlining an industrial software pipeline, with the example of a widely used use case, that respects both RTP and RTE requirements.

## 1 Introduction

Deep learning has achieved massive success over the past decade in several domains and high stakes (Dong et al., 2021). It has been a cornerstone ever since in almost all aspects of scientific discoveries. However, it is to be noted that deep learning, although predominant in today's scientific understanding, comes with its inherent shortcomings (Talaei Khoei et al., 2023; Saeed and Omlin, 2023; Boulemtafes et al., 2020). In this paper, we shall investigate the intricacies of two such pivotal shortcomings: **Privacy** and **Explainability**. We shall thereafter substantiate our findings with a famous use case from previous studies.

Firstly, the deep models, although achieve superior performance across domains, are inherently prone to sensitive data leakage (Boulemtafes et al., 2020). It has been extensively shown that the models can leak information about the data with which it has been trained. Even, especially in the biomedical domain, it has been shown that simple 'linkage' attacks can exploit anonymized electronic health records (Sweeney, 2015). Due to the inherently vulnerable nature of deep models, it is not only prone to severe privacy attacks including membership inference (MIA) (Shokri et al., 2017; Hu et al., 2022), model stealing (Oliynyk et al., 2023), model inversion (Fredrikson et al., 2015; Wang et al., 2021; Veale et al., 2018) etc but also impose a threat on the applicability of deep learning in real-world applications.

Secondly, the deep models are *hard* to explain (Saeed and Omlin, 2023). Neural nets are nonlinear systems, often try to learn the (approximate) distribution of the training data. However, once a model is deployed in an *open-world* setting (Zhu et al., 2024), there is no gold label to cross-check the prediction obtained as a result; the subtle notion of *trust* on the prediction is pivotal to move forward with the same. But as these models are excessively critical, how the model comes to the prediction is a non-trivial phenomenon (Jentzen et al., 2023).

To tackle the first problem, researchers have developed several strategies for privacy-preserving machine learning (Boulemtafes et al., 2020). This includes homomorphic encryption (Pulido-Gaytan et al., 2020), PATE (Papernot et al., 2018), differential privacy (DP) (Abadi et al., 2016) etc; depending on the setting and motivation, we incorporate these methods to make the model *robust* against diverse privacy attacks. However, differential privacy (DP), over the years, has been established as a *gold-standard* for privacy-preserving machine learning (Blanco-Justicia et al., 2022; Suriyakumar et al., 2021) mainly due to its *worst case guarantee* against *any* inference attacks (Abadi et al., 2016; Dwork and Roth, 2014). Furthermore, other types of privacy attacks are not as mature and/or successful as MIA is (Rigaki and Garcia, 2023). In this paper, we have specifically worked on DP models[1] with diverse *privacy guarantee*s aka $\epsilon$ (Dwork and Roth, 2014).

On the other hand, to tackle the explainability problem, researchers have developed an array of methods to audit the model decisions. This includes (but not limited to) local and global explainability methods, inherently interpretable machines, etc (Saeed and Omlin, 2023). Among these, local post-hoc explainers have emerged as popular and widely adopted tools for model auditing in recent times due to their 'plug-and-play' nature (Bhatt et al., 2020). In this work, we employ five popular post-hoc explainers relevant to our use case to obtain explanations.

Although privacy and explainability aspects have been vastly explored independently, there is little to no work incorporating them together. Despite the rapid advancements in AI, integrating privacy and explainability in high-stakes domains remains an unsolved and pressing challenge. Existing research has treated privacy and explainability as separate challenges, leaving a significant gap in understanding their interplay. This gap becomes critical in high-stakes applications where both are non-negotiable. Researchers in privacy-preserving machine learning have investigated several aspects of differential privacy, such as the scalability of DP models (Beltran et al., 2024), their fairness (Fioretto et al., 2022), robustness (Tursynbek et al., 2020), privacy-utility trade-off (Zheng et al., 2024), etc. On the other side, researchers have worked on different aspects of explainable AI (XAI) such as faithfulness (Lyu et al., 2024), robustness (Alvarez-Melis and Jaakkola, 2018), and the quality of explanations (Zhou et al., 2021) to name a few. In this paper, we take the first comprehensive step toward bridging the divide between privacy-preserving models and explainable AI in high-stakes domains. We explore the unique challenges that emerge when attempting to integrate privacy and explainability aspects in high-stakes applications. We investigate the underlying causes of these challenges, examine the trade-offs involved, and discuss key considerations necessary for developing frameworks to successfully incorporate these two critical aspects effectively.

Integrating privacy and explainability is non-trivial, but addressing these issues is crucial for ensuring trustworthy and effective AI systems in high-stakes. In this context, we argue that it is equally important to identify use cases where achieving 'the best of both worlds' is not just desirable but a *requirement*. For example, if the training data is open source or an interpretable model (e.g. decision trees) is considered for the study, our research may not be well-motivated or the findings may not be 'worthy'. This is why, for this paper, we have considered the well-established use-case of disease detection from chest x-ray (Al-qaness et al., 2024). The reason to choose the same is that medical records are (almost) always subject to privacy preservation, and it is of paramount importance that, with explanations, the stakeholders including both physicians and patients, can *trust* the model predictions. Post-hoc explainers, which we use in this paper, have been extensively utilized for our chosen and similar use cases in past (E. Ihongbe et al., 2024; Saxena et al., 2022; Ifty et al., 2024).

The dual mandates of the Right-to-Privacy (RTP) (Thomson, 1975) and the Right-to-Explanation (RTE) (Vredenburgh, 2022), increasingly imposed by governments for high-stakes AI applications, underscore the urgency of addressing these intertwined imperatives. Despite their critical importance, prior research lacks a comprehensive and systematic feasibility study that explores the practical integration of these rights within AI systems. Fundamental questions remain unanswered: How should we evaluate the quality of explanations for DP models? Can existing popular local post-hoc explainers reliably function in this setting, or do they falter under the constraints of differential privacy? If they fail, what alternative methods can generate comparable private explanations? Our work makes a fundamental contribution by systematically addressing

---

[1]Throughout the paper we use DP model(s)/explanation(s) and private model(s)/explanation(s) interchangeably.

these pressing questions, providing not only foundational clarity but also actionable insights grounded in rigorous empirical analysis. Formally, our central research question in this paper is:

■ Do DP models and post-hoc explainers *go together*?

Our contributions are as follows.

- We propose the desiderata for private explanations to follow;

- We investigate the interplay between the DP models and post-hoc explainers, subject to the desiderata proposed;

- We conduct extensive experiments and report our findings: we consider three types of widely used CNN models, six distinct $\epsilon$ values to train them on, and five popular post-hoc methods for our experiment;

- We present a rigorous study on the mechanistic interpretation of the DP models, depicting the interplay of the post-hoc explainers with DP models;

- We propose a novel framework that can achieve RTP and RTE together for high-stakes applications; we exemplify the same by outlining an end-to-end privacy-preserving pipeline for our use case.

The rest of the paper is organized as follows. We brief the preliminary concepts of DP and post-hoc explainers in Section 2. We point out the potential privacy breaches, followed by an extensive discussion on several quantifiable notions for explanations' quality, and their inherent shortcomings in Section 3. We formally introduce our proposed postulate and quantifiable measures for the same in Section 4. We detail our experimental setup in Section 5. We discuss our findings in Section 6. We mechanistically interpret the models to explain our findings in Section 7. Next, we explore Local Differential Privacy and its applicability in our context, in Section 8. We outline our novel software pipeline to *reconcile* privacy and explainability in Section 9. We present the related works in Section 10. Finally, we conclude in Section 11.

## 2 Preliminaries

**Privacy-preserving machine learning** is the building block for RTP in modern-day ML systems. Researchers have developed numerous strategies for privacy-preserving machine learning, including homomorphic encryption, PATE, and differential privacy (DP) (Boulemtafes et al., 2020; Papernot et al., 2018; Abadi et al., 2016). Among these, DP has emerged as the *gold standard* due to its strong *worst-case guarantee* against inference attacks (Blanco-Justicia et al., 2022; Dwork and Roth, 2014). In this work, we focus on DP models with varying privacy guarantees.

A randomized mechanism $\mathcal{F}$ satisfies $(\epsilon, \delta)$-DP if, for any two neighboring datasets $D$ and $D'$ differing in at most one record, and for all measurable subsets $\mathcal{S}$ of the output space,

$$\Pr[\mathcal{F}(D) \in \mathcal{S}] \leq e^{\epsilon} \Pr[\mathcal{F}(D') \in \mathcal{S}] + \delta.$$

Here, $\delta$ represents the probability of the mechanism failing to provide privacy guarantees. Specifically, it accounts for the small chance that the added noise does not sufficiently obscure the presence or absence of an individual in the dataset. Smaller values of $\delta$ are preferred.

DP comes with a few interesting properties such as sequential composition, parallel composition and, post-processing. In our study, post-processing is most relevant which lets the user perform arbitrary operations on the output of a DP mechanism without hampering its privacy (Dwork and Roth, 2014).

To make a model differentially private (DP), it undergoes privacy-preserving training, with DP-SGD being the most prominent method (Ponomareva et al., 2023). It introduces Gradient clipping ensuring bounded contributions of individual data points and noise to the gradients during each update to ensure differential privacy (Abadi et al., 2016). Gaussian noise is commonly used due to its well-analyzed privacy guarantees.

Alternatively, laplacian noise is employed in some contexts where bounded sensitivity is easier to calculate (Dwork and Roth, 2014). For DP-SGD, Gaussian noise is predominantly used, whereas Laplacian noise is popular in Local Differential Privacy (LDP) (Dwork and Roth, 2014; Ponomareva et al., 2023).

Local Differential Privacy (LDP) provides privacy guarantees at the data source. A mechanism $\mathcal{F}$ satisfies $\epsilon$-LDP if, for any two inputs $x$ and $x'$ and all outputs $y$,

$$\Pr[\mathcal{F}(x) = y] \leq e^{\epsilon} \Pr[\mathcal{F}(x') = y].$$

In this work, we denote the non-private model as $\mathcal{M}$. We obtain its private counterpart $\mathcal{M}'$ by retraining with DP-SGD.

**Explainable machine learning** is crucial for Right-to-Explainability (RTE). As mentioned earlier, this paper focuses on local, post-hoc explainers (denoted as $\mathcal{I}$). Broadly, these explainers can be classified into two categories: Perturbation-based methods and Gradient-based methods. Both types of methods take a model ($g$) and a data point ($x$) as input and output a feature attribution score(s) against $g(x)$. A feature attribution score (FAS) (or a feature attribution vector) consists of scores assigned to individual features of the given input representing their importance towards the classification the given model comes up with. A positive attribution score implies a feature has positively contributed towards the classification and a negative attribution score shows that a feature does not positively contribute towards the classification. Given an explainer and a data point ($x$), we denote the FAS which explains the prediction of $\mathcal{M}$ and $\mathcal{M}'$ as $s$ and $s'$ respectively; given $\mathcal{M}(x) = \mathcal{M}'(x)$.

In our use case, $s$ (and $s'$) are typically computed on a per-pixel or per-element basis, and they generally match the dimensions of the input or the corresponding intermediate layer. In the next section, we shall discuss the non-private setup and potential security breaches involved.

## 3 Two Side of the Coin: Privacy and Explainability

### 3.1 The Privacy Aspect

The non-private setup consists of an adversary **A**, having access to a data-point $x$, a non-private model $\mathcal{M}$, the output vector **V** obtained from $\mathcal{M}$ for $x$, an explainer $\mathcal{I}$ generating a feature attribution score $s$ against $\mathcal{M}(x)$.

From the point of **A**, we identify two distinct ways of attacking and leaking the information from the trained model[2]:

1. For a given data point $x$, assuming **A** has access to the output vector **V** from the trained model $\mathcal{M}$, **A** can perform an MIA on the model (Hu et al., 2022).

2. Assuming **A** has access to the feature attribution score $s$ of $x$ generated by $\mathcal{I}$ subject to $\mathcal{M}(x)$, **A** can leverage $s$ to execute an MIA on $\mathcal{M}$ (Shokri et al., 2021), especially when $\mathcal{I}$ is *faithful*.

However, as $\mathcal{M}'$ is the DP counterpart of $\mathcal{M}$, it inherits the post-processing property (Dwork and Roth, 2014) which makes *any* mechanism including obtaining **V**, $s'$, from $\mathcal{M}'$ to produce results which are also DP, *i.e.*, *any* post-hoc explanations for an $(\epsilon, \delta)$-DP model also satisfy $(\epsilon, \delta)$-DP for the training dataset (Patel et al., 2022). Thus, using $\mathcal{M}'$ we can mitigate both of the breaches.

### 3.2 The Explainability Aspect

As discussed earlier, in this study, we are exclusively considering local post-hoc explainers (Lundberg et al., 2020; Huber et al., 2021; Lundberg and Lee, 2017). The *quality* of these local explanations are judged across

---

[2]As in this paper we are working with DP models, we have considered MIA as the only type of potential privacy attack for simplicity. However, it has been shown that other classes of privacy attacks (model stealing, model inversion, etc) are also very much possible with the two ways of attacking mentioned, and DP potentially can safeguard against a few other attacks besides MIA (Rigaki and Garcia, 2023). However, these fall beyond the scope of the present study.

several parameters (Hedström et al., 2023): `Faithfulness` (Lyu et al., 2024), `Robustness` (Alvarez-Melis and Jaakkola, 2018), `Localization` (Zhang et al., 2018; Theiner et al., 2022), `Complexity` (Chalasani et al., 2020; Nguyen and Martínez, 2020), `Randomization` (Adebayo et al., 2018), are mostly accentuated in previous studies.

Unequivocally, `faithfulness` is the most crucial among all (Lyu et al., 2024). `Faithfulness` is loosely defined as how well the explainer reflects the underlying reasoning of the model and has been extensively quantified in diverse ways (Li et al., 2023; Hedström et al., 2023): `Sufficiency (SF)` (Dasgupta et al., 2022), `Infidelity (IF)` (Yeh et al., 2019), `Insertion/Detection AUC (I/D-AUC)` (Petsiuk et al., 2018), `Pixel Flipping (PF)` (Bach et al., 2015), `Iterative Removal of Features (IROF)` (Rieger and Hansen, 2020), `Ordered Perturbation` based metrics `(OPs)` (Samek et al., 2016) are to mention a few. However, the evaluation metric(s) are not necessarily flawless, which we're going to discuss next.

### 3.3 Pitfalls of the Evaluation Metrics

Firstly, in the perturbation-based metrics such as `PF`, `I/D-AUC`, `IROF`, `IF` etc while applying operations such as flipping the pixels, inserting and/or deleting features, performing `meaningful perturbation` on the input space to generate synthetic inputs as a part of their evaluation process, do not crosscheck whether the generated input is out-of-distribution with respect to the trained model (Hase et al., 2021; Chang et al., 2018). Secondly, previous studies have extensively shown that test-time input ablation is often prone not only to generating out-of-distribution (OOD) synthetic input (Hase et al., 2021; Haug et al., 2021) but also are *socially misaligned* (Jacovi and Goldberg, 2021). Thus, metrics for example `I/D-AUC`, `IROF` etc are suspected to be severely misleading; even metrics such as `SF` are also substantially constrained and do not provide a universal overview of `faithfulness`. Thirdly, metrics such as `OPs` are often excessively similar to the mechanics of the explainer itself. Instead of evaluating `faithfulness`, these methods primarily compute the similarity between the evaluation metric and explanation techniques, assuming the evaluation metric itself to be the ground truth (Ju et al., 2021). Li et al. (Li et al., 2023) has acknowledged the same in their benchmark $\mathcal{M}^4$ that LIME (Ribeiro et al., 2016) and `OPs` are methodologically similar thus, the evaluation maybe skewed. Lastly, all these evaluations are based on naive assumptions (e.g.: erasure (Jacovi and Goldberg, 2020)), derived from a set of seemingly valid observations. Therefore, these quantitative metrics are not necessarily axiomatically valid. Even axiomatic necessary tests such as `Model Parameter Randomization test` (Adebayo et al., 2018) happen to have a set of empirical confounders (Yona and Greenfeld, 2021; Kokhlikyan et al., 2021; Bora et al., 2024). To the best of our knowledge, there has not been any universally valid necessary and sufficient approach for faithfulness evaluation (Lyu et al., 2024).

Furthermore, the absence of ground truth for models' reasoning makes it an open challenge to universally quantify faithfulness and evaluate a faithfulness measure precisely, i.e. when a universal ground truth for reasoning is not available, axiomatically quantifying how much a faithfulness measure is more reliable than others is challenging; however, in the existing literature, faithfulness measures evolve by addressing the discovered shortcomings of their predecessors (Lyu et al., 2024). This also leads to the well-known disagreement problem (Krishna et al.): even if explainers are `faithful`, their inherent mechanisms for calculating feature importance can lead to different explanations. In the subsequent section(s), we'll try to figure out whether we can bypass these limitations for our study.

### 3.4 Aspiration for the Alternatives

Acknowledging these inherent limitations in current evaluation methods, we propose two key remarks:

1. **Remark 1.** Expert oversight should determine whether an explanation is *suitable* for high-stakes applications.

2. **Remark 2.** Explanations should align with local constraints and contexts, even when (so-called) *faithfulness* cannot be measured reliably.

To address the first requirement, in our use case we ensure that concerned physicians first receive X-ray images along with predictions and explanations. Results are only communicated to patients after the physician has completed a formal review and certified both the prediction and the explanation.

For the second remark, we are introducing the `localization assumption` (LA) and quantifying the same with a class of measures we collectively named the Privacy Invariance Score (PIS) for explanations.

## 4 Localization Assumption & Privacy Invariance Score

Right-to-Privacy (RTP) (Thomson, 1975) and Right-to-Explanation (RTE) (Vredenburgh, 2022) are two inalienable aspects of modern-day ML software. We have already mentioned that previous works have shown potential security breaches leveraging the explanation in Section 3.1. So, RTE can hamper RTP but is the converse true? If yes, how can we quantify the same? We first propose the desiderata for explanations in this setting and then run extensive experiments to quantitatively *judge* the explainers.

### 4.1 Description of the Setting

It is shown that adversaries can leverage *sensitive* information from explanations but due to post-processing of DP, any DP model will always output a private explanation (Dwork and Roth, 2014). We want to check the extent to which the explanations from a DP model can be used as a proxy for that of the non-private model here. Formally, for a non-private model $\mathcal{M}$, its private counterpart $\mathcal{M}'$; consider a set of post-hoc explainers $\mathcal{E}$ used for auditing $\mathcal{M}$ and $\mathcal{M}'$. For an input $x \in X$ (where $X$ is the valid input space), each $\mathcal{I} \in \mathcal{E}$ produces explanations $s$ and $s'$ of $x$ for $\mathcal{M}(x)$ and $\mathcal{M}'(x)$ respectively, Privacy Invariance Score **(PIS)** over a given tuple $(\mathcal{M}, \mathcal{M}', x, \mathcal{I})$ is defined as follows.

**Definition 1.** Privacy Invariance Score **(PIS)**: Given a tuple $(\mathcal{M}, \mathcal{M}', x, \mathcal{I})$, given $\mathcal{M}(x) = \mathcal{M}'(x)$, PIS is defined as $sim(s, s')$, where $sim(\cdot, \cdot)$ is a *similarity* measure, and $s = \mathcal{I}(\mathcal{M}, x)$ and $s' = \mathcal{I}(\mathcal{M}', x)$.

### 4.2 Localization Assumption

As discussed earlier, different works evaluated explainers in various ways. These ad-hoc norms are often unique to each paper and inconsistent (Jacovi and Goldberg, 2020). However, practitioners have proposed some necessary axiomatic desiderata (Lyu et al., 2024) such as the well-established `Implementation Invariance (II)` criterion proposed by Sundararajan et al. (Sundararajan et al., 2017).

According to `II` if $\mathcal{I}$ is *faithful* and $\mathcal{M}(x) = \mathcal{M}'(x) \ \forall x \in X$, i.e. $\mathcal{M}, \mathcal{M}'$ are *functionally equivalent* (FE) then $s = s' \ \forall x$. Commonly used explainers like `Integrated Gradient` (Sundararajan et al., 2017), `SmoothGrad` (Smilkov et al., 2017), `DeepLift` (Li et al., 2021), `layerwise relevance propagation (LRP)` (Montavon et al., 2019) etc had been extensively accessed based on `II` (Sundararajan et al., 2017).

Following these lines of prior explainability research, Jacovi et al. later formally proposed `The Model Assumption (MA)` (Jacovi and Goldberg, 2020):

*"Two models will make the same predictions if and only if they use the same reasoning process."*

Nevertheless, in our setting, the accuracy of $\mathcal{M}'$ is generally less (Abadi et al., 2016), and we cannot say $\mathcal{M}$ and $\mathcal{M}'$ are FE (Sundararajan et al., 2017). Furthermore, as $X$ in practice can be arbitrarily large (theoretically could be countably infinite), quantifying whether an explainer ($\mathcal{I}$) is *sufficiently* trustworthy or not is often impractical. Also, finding a counterexample that violates the condition implying $\mathcal{I}$ is not faithful is computationally expensive. Hence, we modify `MA` and propose the `localization assumption` for evaluating the explainers' quality in our setting.

First of all, in both the `II` and `MA` we do not advocate comparing any arbitrary models trained on the same data, having the same $X$ to compare. For example, for a finite $X$, a trained neural network and a decision tree could have the same prediction $\forall x \in X$. That doesn't mean their *reasoning* is similar as the algorithms themselves are different. However, the private model $\mathcal{M}'$, in our setting, has identical architecture to that of $\mathcal{M}$. The fundamental goal of differential privacy is assumed to be masking individual contributions of

the training set rather than completely changing its overall reasoning. However, as noise is induced in the gradient during the training, the parameters are expected not to remain entirely the same in the private model. Therefore, we assume that their reasoning should primarily be *similar*, if not the *same*. Formally, our adapted version of the MA is:

**Assumption 1 The `Localization Assumption (LA)`.** *For a given tuple* $(\mathcal{M}, \mathcal{M}', x, \mathcal{I})$ *having* $\mathcal{M}(x) = \mathcal{M}'(x)$, $sim(s, s') >= \theta$. *Where* $\theta$ *is a predefined similarity threshold.*[3]

In this context, since we are using a proxy model (in our case, a DP model) as a substitute for the original model (the non-private model), we argue that beyond satisfying LA, the system must also meet two additional requirements:

1. **Performance comparability** (*Perf Comp.*): The proxy model should achieve comparable performance metrics to the original model. For this study, we focused on accuracy (acc) since in our test set, we put equal weightage on all classes and there is no class imbalance (further details are provided in Section 6). Specifically, we report $Acc_{\mathcal{M}'/\mathcal{M}} = \frac{\text{acc. of } \mathcal{M}'}{\text{acc. of } \mathcal{M}}$ for our experiments.

2. **Alignment with the original model** (*Alignment*): The proxy model must closely align with the original model in its predictions. In our study, we measure the same with the agreement on the "hard predictions" obtained from $\mathcal{M}$ and $\mathcal{M}'$ over the test set. (Dis)agreement has been extensively used in previous studies due to its simplicity and interpretability (Klabunde et al., 2025).

These two requirements are critical for ensuring the effectiveness of the proxy model. *Perf Comp.* is necessary because a significant drop in the proxy model's performance would undermine its utility as a stand-in for the original model, especially in high-stakes applications. *Alignment* is equally important because, without sufficient agreement on predictions, the proxy model will fail to replicate the original model's decision-making patterns, making it ineffective as a proxy. We refer to a proxy model that meets both these requirements as *functionally comparable* w.r.t. the original model.

Next, we investigate how we can measure the 'similarity' described in the `Localization Assumption`.

### 4.3 The Notion of Similarity

As previously discussed in PIS, we aim to measure the similarity between pairs of explanations; it is essential to account for two key factors here: the context of comparison and human understanding of that comparison. In this framework, given $(\mathcal{M}, \mathcal{M}', x, \mathcal{I})$ and $\mathcal{M}(x) = \mathcal{M}'(x)$, we seek to investigate two primary aspects:

- To what extent do $s$ and $s'$ *agree*?
- Wherever they agree, what is the degree of that agreement?

Firstly, explanations typically include both positive and negative attributions, where positive attributions favor the classification, and negative ones do not. To evaluate two explanations' similarity, we first compute a disagreement score (DS, measured in %), by measuring the mismatch of corresponding attribution type (negative or positive) in $(s, s')$. DS is a (crude) measure for assessing how much the reasoning of the models, as indicated by $\mathcal{I}$, diverges at the pixel or element level. In several highstakes (e.g. biomedical), as both position and attribution are important, we employ DS primarily as a sanity check in our case study with a threshold (hyperparameter) of 15%.

Secondly, after measuring DS and eliminating the explainers not adhering the threshold, we assess the attribution correlation for the rest. We start by segregating the positive attributions from the negative ones, as the former contribute positively to the classification, while the later may (partially or completely) contribute to other classes or contribute negatively to the output class or a combination of these two (Ancona et al., 2017; Samek et al., 2017). Consequently, we compute the Kendall's tau between pairwise positive pixels of $s$ and $s'$ to measure PIS. While alternative notions for similarity like cosine similarity or $L_P$ norms

---

[3]It is worth nothing that LA and `Localization` discussed in 3.2 are different.

exist, cosine is unsuitable in high dimensions due to near-orthogonality of random vectors (Mohammadi and Petridis, 2022; Wyner, 1967), and $L_P$ norms require normalization, losing one degree of freedom unlike correlation.

## 5 Experimental Setup

### 5.1 Description of the Dataset

Our dataset comprises 2,000 Pneumonia cases sourced from the Chest X-ray dataset by (Patel, 2020), and 2,000 TB cases randomly sampled from the NIAID TB Portal Program dataset (National Institute of Allergy and Infectious Diseases). To create the 'Normal' subset, we include an equal split of 1,000 unaffected Pneumonia samples from the unaffected class in the aforementioned Chest X-ray dataset (Patel, 2020) and 1,000 unaffected TB samples from Rahman et al. (Rahman et al., 2020), totaling 2,000 normal cases. For evaluation, the test set contains 200 images from each class.

This paper, unlike prior studies, is not focused on the empirical studies of private models in healthcare (Naresh et al., 2023; Khalid et al., 2023). Instead, we concentrate on the applicability of commonly used explainers in a privacy-preserving environment. Our aim is to investigate whether RTE and RTP can be achieved simultaneously. As a result, rather than a large dataset where extensive experimentation on several types of CNNs with DP under a closely controlled environment is severely challenging and computationally demanding (Ponomareva et al., 2023), we focus on first making a dataset of an appropriate size where we have considered the commonly available and utilized disease class from the previous studies. However, after our primary experiment, we, within the limits of feasibility, experimented with another benchmark dataset: CIFAR-10, and we arrived at a similar conclusion. Details can be found in Appendix A.

### 5.2 Choosing the Privacy Budget ($\epsilon$)

Selecting $\epsilon$ is critical to balancing privacy requirements and utility. The theoretical school advocates for small values of $\epsilon$ (e.g. $\epsilon \leq 1$), offering strong worst-case guarantees (for e.g. $\epsilon = 0.1$ *theoretically* limits MIA success rates to 52.5%). In contrast, industrial settings often adopt large $\epsilon$ (e.g. $\epsilon > 7$), which, while practical, *theoretically* yield MIA success rate exceeding 99%.

This dichotomy arises from differing assumptions (Lowy et al., 2024): the theoretical school presumes adversaries with **near-complete knowledge** of training data and **uniform privacy guarantees** for all data points. However, recently Lowy et al. (Lowy et al., 2024) highlighted the impracticality of these assumptions and showed that even $\epsilon \geq 7$ can provide adequate defense against existing MIAs.

While this study does not propose guidelines for choosing $\epsilon$ in biomedical applications, we adopt Google's three-tier framework (Ponomareva et al., 2023):
**Tier 1**: $\epsilon \leq 1$: strong theoretical guarantees, low utility;
**Tier 2**: $\epsilon \leq 10$: practical trade-offs, google advocates this tier primarily;
**Tier 3**: $\epsilon > 10$: high vulnerability, thus excluded from this study.

Our analysis examines $\epsilon \in \{0.4, 0.7, 1, 4, 7, 10\}$, spanning tier 1 and tier 2, to comprehensively evaluate privacy-utility trade-offs.

### 5.3 Description of the Explainers

We primarily selected gradient-based explainers, which are predominantly used in computer vision tasks (Buhrmester et al., 2021) — specifically, `Saliency` (Simonyan et al., 2013), `SmoothGrad` (Smilkov et al., 2017), `Integrated Gradients` (Sundararajan et al., 2017), `Grad-Shap` (Lundberg and Lee, 2017) and `Grad-CAM` (Selvaraju et al., 2019) (details are in Appendix B). These gradient-based explainers rely on the sensitivity the model shows to the features subject to the output it obtains. In other words, these explainers leverage the gradient of the output (or any selected class of interest) w.r.t. the input features[4]. For example, `Saliency` computes the partial derivatives of the output w.r.t. the input given,

---

[4]it can either directly be the element(s) of the given input or the intermediate ones coming from a layer inside the model.

`Integrated Gradients` computes the average gradient while the input varies along a linear path from a baseline. `SmoothGrad` involves adding noise to the input image and generating multiple saliency maps, each corresponding to a slightly different noisy version of the input image; the saliency maps are then averaged to produce a 'smoothed' saliency map. `Grad-CAM` also involves taking gradients of the target output with respect to the given layer. From perturbation-based explainers, we included only `SHAP` (Lundberg and Lee, 2017), specifically `Grad-Shap`. `Grad-Shap`, in a sense, can be viewed as an approximation of `Integrated Gradients` by computing the expectations of gradients for different baselines. Core perturbation methods are less widely adopted for computer vision, and they tend to disobey the "data manifold hypothesis" (Fefferman et al., 2016), as described in the next paragraph. Furthermore, perturbation-based methods like `LIME` (Ribeiro et al., 2016) assign weights to super-pixels rather than individual pixels. As a result, we cannot directly compare the same with other methods we chose, which output per-pixel or per-element attribution values, `Occlusion` can lead to OOD artifacts and is not widely used (Chang et al., 2018). Additionally, some other explainers, such as `DeepLIFT, LRP` disregard even basic faithfulness tests (Sundararajan et al., 2017).

We did not consider model-agnostic perturbation-based explainers (e.g., KernelSHAP, LIME, Occlusion) due to their relatively limited adaptation in computer vision tasks and several criticisms. Firstly, these are known to disobey the "data manifold hypothesis" (Fefferman et al., 2016): Kumar et al. (Kumar et al., 2020) and Sundararajan et al. (Sundararajan and Najmi, 2020) showed these methods often operate outside the data's true distribution. Furthermore, Shokri et al. (Shokri et al., 2021) acknowledged the fact that "*little is known on how models generally perform outside of the data manifold. In fact, it is not even clear how one would measure performance of a model on points outside of the training data distribution: they do not have any natural labels...* ". To the best of our knowledge, not only vanilla models but also how their DP counterparts behave outside the data manifold is unknown. In fact, how different the manifold of DP models is, compared to its non-DP counterpart, and how these perturbation-based explainers may behave subject to DP models' decision boundaries and manifolds, has not been extensively studied prior. Therefore, whether, in an ad-hoc manner, we can even employ these explainers to DP models or the suitability of these explainers in the context of DP models is not clear to us, and we'd like to leave this as an open research question to the community. However, as mentioned earlier, we considered `Grad-Shap` as it closely resembles to `Integrated Gradients`.

### 5.4 Description of the Networks

Disease detection using chest X-ray is a well-regarded case study. In previous works (Al-qaness et al., 2024), the choice of CNNs is significantly prominent, and in this work, we have also chosen the famous CNNs that have been used in previous research. Particularly, We selected ResNet-34 (He et al., 2016), EfficientNet-v2 ('small' version) (Tan and Le, 2021), and DenseNet-121 (Huang et al., 2017) for our analysis due to their competitive performance. Importantly, we ensured that the DP counterparts of these models were *functionally comparable* across a wide range of values for $\epsilon$, to the best of our ability. However, previous studies have shown that DP doesn't make the model learn all classes in a comparable fashion (Suriyakumar et al., 2021; Fioretto et al., 2022) as a result, we report $Acc_{\mathcal{M'}/\mathcal{M}}$ as a measure of *Perf Comp.* and agreement on "hard prediction" to measure *Alignment* on a class-wise basis, allowing for a more nuanced comparison. We detailed our experimental setup in Appendix C.

| | TB Acc. (%) | Pneumonia Acc. (%) | Normal Acc. (%) | Overall Acc. (%) |
|---|---|---|---|---|
| DenseNet | 95.0 | 97.5 | 97.2 | 96.55 |
| EfficientNet | 93.5 | 96.2 | 96.0 | 95.57 |
| ResNet | 93.2 | 96.0 | 96.3 | 95.51 |

Table 1: Acc. Table for models.

## 6 Key Observation

In this section, we present our experimental results to find out whether the post-hoc explainers agree with the `Localization Assumption` (LA), defined in Section 4.2. For `Integrated Gradients` and `Grad-Shap`,

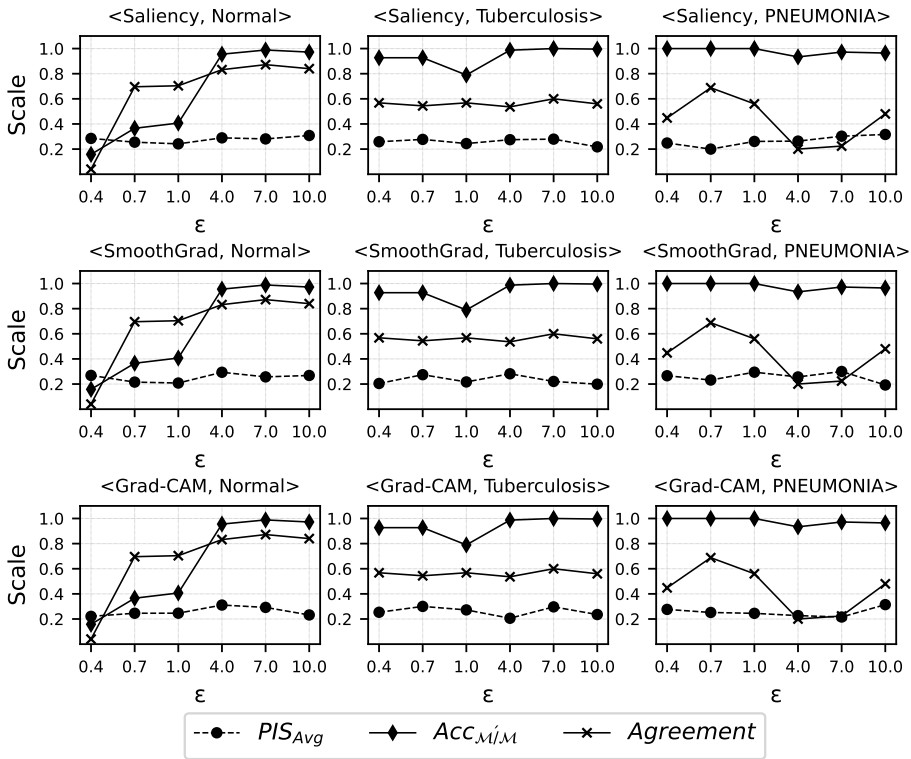

Figure 1: Performance of explainers (ResNet-34)

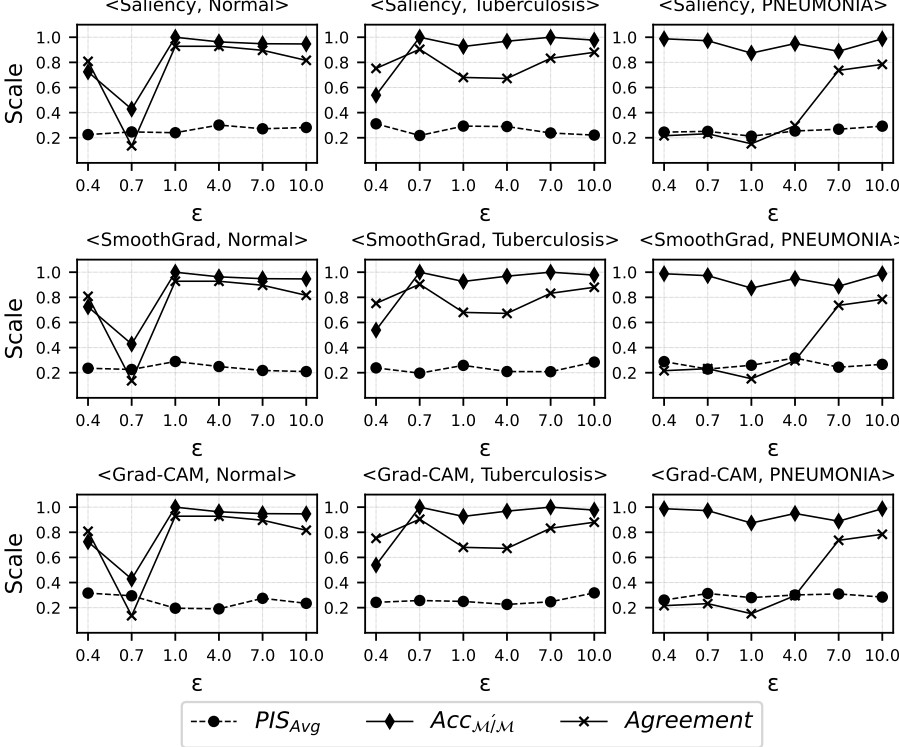

Figure 2: Performance of explainers (DenseNet-121)

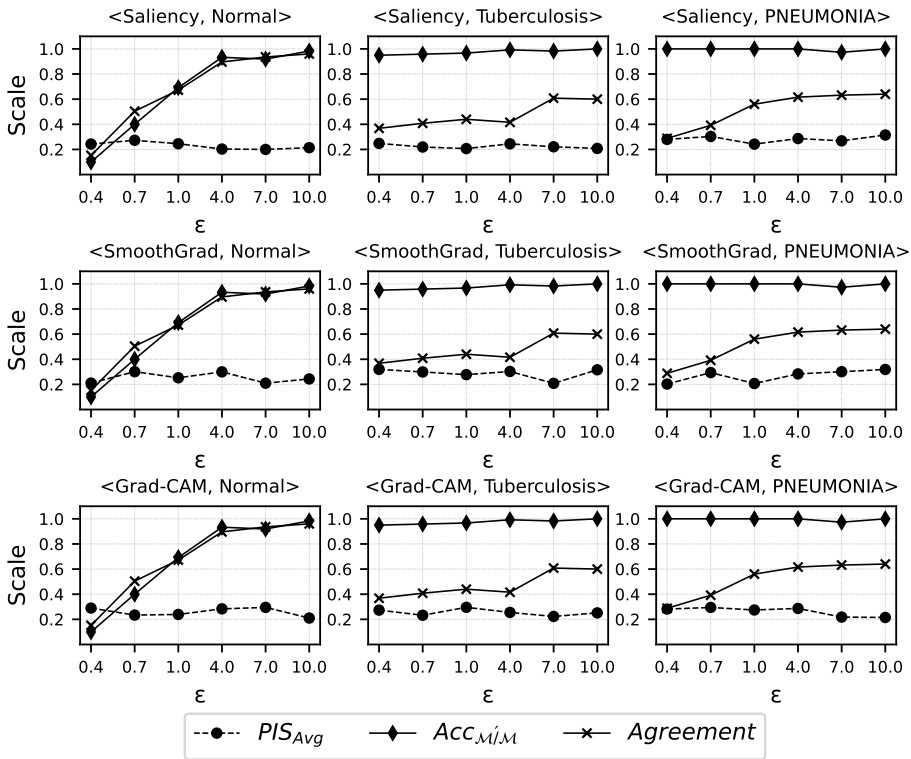

Figure 3: Performance of explainers (EfficientNet-V2)

across all models and test samples chosen, we obtained DS of more than 45%, which directly violates our DS threshold of 15%. As we consider DS as a sanity check for LA, and these two explainers fail this test, we exclude them from further discussions. Across models, for around $30 - 40\%$ of all test samples, `Grad-Cam` explanations violated the DS threshold of 15%; consequently, we have considered the rest of the test samples to calculate PIS for `Grad-Cam`. All test samples passed the DS test across models for `Saliency` and `SmoothGrad`.

We report our findings for the three networks: DenseNet-121, ResNet-34, and EfficientNet-v2 in Figure 2, 1, and 3 respectively. We report class-wise results for each of these networks against each of the three explainers that pass the DS test. Against each $\epsilon$, we report average of the PIS values over the test set ($PIS_{Avg}$), $Acc_{\mathcal{M}'/\mathcal{M}}$, and agreement on the "hard predictions" obtained from $\mathcal{M}$ and $\mathcal{M}'$ over the test set ($Agreement$). $PIS_{Avg}$ gives the extent to which an explainer agrees with the LA in a particular setup; $Acc_{\mathcal{M}'/\mathcal{M}}$ quantifies the *Perf Comp.* requirement, and *Agreement* quantifies the *Alignment* requirement for the private model to be *functionally comparable* w.r.t. the non-private counterpart, defined in Section 4.2. We observe all of the explainers across $\epsilon$, classes, and models are performing very poorly in terms of $PIS_{Avg}$[5]. In fact, the $PIS_{Avg}$ never crosses 0.32; such a low $PIS_{Avg}$ indicates that none of the explainers agree with the LA. In almost all cases, except for the Normal class against low $\epsilon$ values, $Acc_{\mathcal{M}'/\mathcal{M}} \approx 1$, which indicates that almost all private models achieved near equal accuracy as of their base counterparts. Nearly 70% of cases achieved $Agreement > 0.50$. There is no apparent relation or trend the $PIS_{Avg}$ follows with $Acc_{\mathcal{M}'/\mathcal{M}}$ and *Agreement*. All in all, the explanation quality is consistently poor, no matter what happens to the other parameters. We observe the very same trend for another benchmark dataset we considered (see Appendix A). Empirically, it is clear that all gradient-based explainers disregard LA and thus seem not to be suitable with DP models. With that note, we (partially) answered our primary research question that DP models and off-the-shelf explainers (apparently) **don't go** *well together*. But why is it so? We'll try to find out in

---

[5]One may argue that what if we measure similarity without segregating attributes? This, although disregards our philosophy on similarly measure, still we did, and to be precise, never found $PIS_{Avg} > 0.16$.

the next section. Also, we'll explore an alternative route to generate private explanations in the subsequent section.

# 7 Why Explainers Performed so *Poorly*?

As noted in the previous section, the explanations from private and non-private models are largely uncorrelated. Private models are trained using DP-SGD, which involves gradient clipping and adding calibrated noise during backpropagation to achieve DP. To understand the divergent behavior of explainers between these models, we first analyze how DP training alters model parameters. We conduct a comparative study on the (layerwise) representations learned by both private and non-private models, followed by a rigorous investigation into the (local) sensitivity of these representations. While the former reveals how DP training modifies the parameter space, the latter explains how these modifications influence the model's reaction to inputs, ultimately affecting explainer behavior. To the best of our knowledge, this is the first study to comprehensively examine the representations learned by DP models and compare them to non-private models. Moreover, the layer-wise sensitivity analysis has not been previously explored for any model, including DP models. We include the implementation details in Appendix C.

## 7.1 Did the Models *Perceive* the Same *Information*?

In its forward pass, a neural network hierarchically transforms the given input into increasingly abstract (and complex) representations. A Representation (also referred to as a feature representation or simply {intermediate} feature) is the set of activations stored layerwise in the network during the forward pass for a given input. We begin our investigation by examining how these representations differ in the private and non-private models, as this reveals how these two different models process and transform the input data across layers. Understanding the alignment of these representations is critical for analyzing how DP training has altered the model's parameters and how these changes influence the model's overall reasoning.

However, since the training processes for DP model is fundamentally different and designed to achieve distinct goals, we first conduct a formal assessment to determine whether any statistical dependence exists between their layerwise representations. This step ensures that the representations are comparable before further analysis. We denote the representation at layer $l$ as $\sigma_l$.

Statistical tests for activations are often non-trivial and subject to a few constraints such as invariance to permutation of the neurons, orthogonal transformation (Klabunde et al., 2025) etc. Consequently, we use the Hilbert-Schmidt Independence Criterion (HSIC) (Gretton et al., 2007) which operates over a Reproducing Kernel Hilbert Space (RKHS) and can detect dependencies across complex, high-dimensional variables while respecting the aforesaid constraints. We employ the HSIC unconditional independence test using the two-parameter $\gamma$ approximation scheme with a p-value cut-off of 0.05 (Gretton et al., 2012). For HSIC, we used both linear and non-linear (RBF) kernels, but obtained similar results.

From our findings, we were **able to reject** the null hypothesis $\mathcal{H}_0$ for almost all layers[6]. In other words, across models, layerwise representations are **not** independent, for all $\epsilon$. Since the representations are sufficiently comparable, we'll now compute their similarity layerwise. However, before a full-fledged comparison, we present a short note on representational similarity (RS).

### 7.1.1 On Representational Similarity

Representational similarity measures compare neural networks by computing similarity between activations of a fixed set of inputs (in our case, the test set) at a given pair of layers. There's been an extensive line of work on this domain, however each of them can be categorized depending on the notion of *similarity* it regards. Each similarity measure follows a set of assumptions. In our paper, we have considered the most regarded set of assumptions and its corresponding similarity measure, namely Centered Kernel Alignment (CKA) (Kornblith et al., 2019; Gretton et al., 2007). CKA is built upon HSIC and bounded within the (closed) interval of $[0, 1]$. However, there are a few (discovered) shortcomings of CKA, as discussed below.

---

[6]For example, across models we never crossed $7 - 10\%$ of the layers for any $\epsilon$ where we accept $\mathcal{H}_0$

The wildly used CKA with the linear kernel, is equivalent to the RV coefficient and it is already shown that the same seldom yields values close to 1 due to one of its inherent shortcomings: although the coefficient is inherently constrained to values between 0 and 1, it *rarely* reaches values near 1 because the denominator is typically much larger relative to its theoretical maximum value (Puccetti, 2022). Also, the less adapted RBF-CKA is highly sensitive to the kernel width and ineffective for small values (Davari et al., 2022).

Furthermore, very recently, Cui et al. (Cui et al., 2022) have discovered that the inter-example (dis)similarity in the representation space works as a confounder. In their words, "*This leads to spuriously high CKAs even between two random neural networks, and counter-intuitive conclusions when comparing CKAs on sets of models trained on different domains ...*". They fixed this problem by regressing out the confounder from the similarity matrices of two representations. Our analysis uses this *deconfounded* version of CKA (dCKA).

In our approach, since both the non-private and private models share the same architecture, we perform a layer-wise comparison between the corresponding layers of the non-private model and its private counterpart(s). However, as the models under consideration are excessively large, reporting Representational Similarity (RS) for each activation layer individually can become overwhelming due to the sheer volume of data. Instead, depending on the number of layers, we grouped the activation layers into 15/17 clusters[7] and reported the median RS for each cluster, as shown in Figures 4, 5, and 6. From our findings, it is evident that any pair of models learn somewhat similar representations. In the initial layers, they learn almost identical representations, but as we progress through the layers, the similarity deteriorates. Interestingly, the last few layers in all model pairs exhibit significant dissimilarity compared to the first few.

All in all, the private models don't perceive the data in a strongly correlated manner w.r.t. their non-private counterparts. However, we argue that this is not the only reason for the explanations to be altered, as explanations are based on the *sensitivity* of the features subject to a class of interest, the model shows. In the second phase of our investigation, we shall look into how much, for an obtained output, the model is *sensitive* to all the layerwise representations it obtained in the forward pass.

## 7.2 Do the Models Show *Similar* Sensitivity?

In this section, we investigate how sensitive the model's output is to the layerwise representations generated during the forward pass. Formally, for a given layer $l$ and a class of interest (here, "hard prediction") $\Theta$, we compute the gradient: $\nabla_{\sigma_l}\Theta$. This gradient is particularly important for two reasons: first, it quantifies the influence of an infinitesimal perturbation in the representations at layer $l$ on the final output. Second, as previously discussed in section 5.3, gradient-based explainers also leverage $\nabla_{\sigma_l}\Theta$ to generate explanations[8]. Thus, explanations, in essence, can be unanimously viewed as the output of a class of *mechanisms* applied to $\nabla_{\sigma_l}\Theta$, subject to specific input(s), layers and/or baselines (wherever required).

This is why the comparability of explanations across models depends on whether $\nabla_{\sigma_l}\Theta$ obtained layerwise are comparable between private and non-private models. Moreover, as the level of abstraction in representations varies across layers, we examined $\nabla_{\sigma_l}\Theta$ at all activation layers to understand the full spectrum of sensitivity across the models.

However, unlike representations, working with $\nabla_{\sigma_l}\Theta$ presents a few more challenges. First, since it directly depends on the model's final output, we need to ensure that both models predict the same class for the input instance in order to make the gradients comparable. If the models predict different classes, the gradients will reflect sensitivities toward those different outputs, making direct comparisons inappropriate. To address this, we could restrict our analysis to the subset of the test set where both models make identical predictions. However, this approach has a distinct problem: RS is calculated over the entire test set, whereas $\nabla_{\sigma_l}\Theta$ would be evaluated over this reduced subset. This mismatch in the sample space (test set) immediately invalidates the direct application of *any* quantitative similarity measures. Conversely, if we use the reduced dataset for evaluating RS, we will not be able to capture the *true* similarity between layers. Additionally, unlike RS, we lack well-established assumptions for defining similarity in the case of $\nabla_{\sigma_l}\Theta$. These inherent challenges make

---

[7]For ResNet-34 and EfficientNet-V2, we considered all 17 `ReLU` and 102 `SiLU` layers respectively, constituting 17 clusters (17|102). For DenseNet-121, we considered all 120 `ReLU` layers, constituting 15 clusters (15|120).

[8]Typically, $l$ refers to the input layer, except for `Grad-CAM`.

it hard to establish a one-to-one correspondence between the similarity of layerwise representations and the similarity of their corresponding sensitivity($\nabla_{\sigma_l}\Theta$).

Due to these inherent challenges, we only conduct hypothesis testing using HSIC, as previously mentioned, to check whether the sensitivity of representations is statistically (in)dependent, where the corresponding outputs match[9]. Here also, we employed both linear and non-linear (RBF) kernels but obtained similar results.

From our findings, we were **unable to reject** the null hypothesis $\mathcal{H}_0$ for almost all layers[10]. In other words, unlike representations, the *sensitivity* of representations is independent across models. Regarding `Grad-CAM`, we typically use the last layer to generate the CAM. Our experiment showed that the gradients of these layers exhibit independence, meaning the CAMs produced by different models will not be comparable. Similarly, other explainers relying on $\nabla_{\sigma_l}\Theta$ cannot produce aligned and substantially comparable explanations, as representations across levels along with the last layer show **independent** sensitivity between non-private and their private counterparts[11].

$\odot$ Overall, neither the DP models extract the features the way a non-private model does, nor can they exhibit the sensitivity over the features in a similar fashion. $\nabla_{\sigma_l}\Theta$ can be viewed as how $\Theta$ changes for a *infinitesimally* small perturbation around $\sigma_l$. We have observed $\sigma_l$ are mildly similar across layers for non-private and private models; however, '$\nabla$' operator being a crude first-order approximation only addresses the immediate, linear response of the output w.r.t perturbations around $\sigma_l$ which, in our case, is **independent** to that of non-private model(s). In other words, the *different* set of parameters obtained with DP training is **sufficient** for the non-private model to show divergent sensitivity across layers, which, in turn, makes the explanations incomparable. DP training was primarily meant to resist MIA w.r.t. the training data, but the training goes much beyond the scope, and drastically alters the overall representation and sensitivity space of a model. Which, as demonstrated, makes the popular post-hoc methods fundamentally nonfunctional. With that note, we now have arrived at a **firm conclusion** that due to DP models' inherent nature, off-the-shelf explainers **don't go** *together* with them.

## 8 Is there any alternative way to have private explanations?

We now know that the DP models cannot accommodate widely used post-hoc explainers primarily due to their own nature. As a result, we cannot move forward with DP models to obtain private explanations that will be useful as a proxy to that of the non-DP counterpart. However, *just* for private explanations, we don't need that. We opt for an *alternative route* to achieve the same *locally*.

In this approach, rather than training the model with DP-SGD, we use the non-private model as usual and take its explanations. We add calibrated noise to the explanation to make it Local Differential Private (LDP).

■ Formally, given a function $f : D \rightarrow \mathbb{R}^d$, the Laplace mechanism $\mathcal{A}$ is defined as:

$$\mathcal{A}(D) = f(D) + \text{Lap}(0|t)^d,$$

where $\mathcal{A}$ satisfies $\epsilon$-differential privacy for $t = \frac{\Delta f}{\epsilon}$, and $\Delta f$ is the sensitivity[12] of $f$ (Dwork and Roth, 2014).

Here, noise is directly added to the heatmap (explanation) obtained from the explainer. Following Fan (Fan, 2018), the sensitivity of a query for an image with $c$ channels and $k$ possible pixel intensities is $(k-1)nc$, where $n$ is the *maximum* number of *differing pixels*. To ensure $\epsilon$-differential privacy, if we apply the Laplace

---

[9]Note, this is the dataset considered for PIS.

[10]In no model did we reject $\mathcal{H}_0$ for more than 3-7% of the layers across $\epsilon$. Notably, for DenseNet-121, all layers were found to be independent.

[11]A natural question arises: could an explainer, despite of consistently incomparable sets of $\nabla_{\sigma_l}\Theta$, produce similar explanations? Based on our experiments, we did not identify any commonly used explainer that exhibited this property for the models selected. Furthermore, we argue on the `faithfulness` of such explainers, if they exist, would be questionable. Conversely, assessing the quality of explanations when (local) sensitivity is consistently comparable across *any* pair of models falls outside the scope of this paper.

[12]This sensitivity is different from the (local) sensitivity we discussed in section 7.

| ε | 1 | 2 | 3 | 4 | 5 | 6 | 7 | 8 | 9 | 10 | 11 | 12 | 13 | 14 | 15 | 16 | 17 |
|---|---|---|---|---|---|---|---|---|---|---|---|---|---|---|---|---|---|
| ε = 10 | 0.97 | 0.88 | 0.86 | 0.82 | 0.66 | 0.63 | 0.64 | 0.64 | 0.64 | 0.64 | 0.65 | 0.64 | 0.64 | 0.64 | 0.66 | 0.66 | 0.66 |
| ε = 7 | 0.98 | 0.91 | 0.90 | 0.82 | 0.60 | 0.55 | 0.55 | 0.55 | 0.57 | 0.59 | 0.59 | 0.59 | 0.58 | 0.57 | 0.58 | 0.58 | 0.61 |
| ε = 4 | 0.97 | 0.91 | 0.91 | 0.82 | 0.57 | 0.51 | 0.48 | 0.49 | 0.49 | 0.49 | 0.47 | 0.46 | 0.44 | 0.43 | 0.42 | 0.41 | 0.42 |
| ε = 1 | 0.96 | 0.88 | 0.88 | 0.83 | 0.65 | 0.61 | 0.60 | 0.59 | 0.59 | 0.61 | 0.62 | 0.62 | 0.62 | 0.61 | 0.62 | 0.60 | 0.64 |
| ε = 0.7 | 0.98 | 0.85 | 0.85 | 0.80 | 0.65 | 0.60 | 0.59 | 0.62 | 0.62 | 0.63 | 0.63 | 0.64 | 0.63 | 0.62 | 0.63 | 0.60 | 0.61 |
| ε = 0.4 | 0.97 | 0.89 | 0.89 | 0.82 | 0.62 | 0.59 | 0.57 | 0.58 | 0.58 | 0.60 | 0.62 | 0.63 | 0.64 | 0.64 | 0.67 | 0.64 | 0.63 |

Figure 4: dCKA heatmaps for ResNet-34

| ε | 1 | 2 | 3 | 4 | 5 | 6 | 7 | 8 | 9 | 10 | 11 | 12 | 13 | 14 | 15 |
|---|---|---|---|---|---|---|---|---|---|---|---|---|---|---|---|
| ε = 10 | 0.94 | 0.87 | 0.83 | 0.75 | 0.76 | 0.72 | 0.69 | 0.71 | 0.70 | 0.69 | 0.71 | 0.71 | 0.71 | 0.71 | 0.71 |
| ε = 7 | 0.93 | 0.84 | 0.82 | 0.75 | 0.73 | 0.67 | 0.67 | 0.66 | 0.64 | 0.66 | 0.67 | 0.59 | 0.58 | 0.57 | 0.58 |
| ε = 4 | 0.91 | 0.84 | 0.81 | 0.75 | 0.76 | 0.71 | 0.68 | 0.71 | 0.67 | 0.71 | 0.73 | 0.73 | 0.70 | 0.70 | 0.69 |
| ε = 1 | 0.93 | 0.85 | 0.82 | 0.75 | 0.74 | 0.69 | 0.66 | 0.66 | 0.64 | 0.67 | 0.67 | 0.61 | 0.61 | 0.62 | 0.62 |
| ε = 0.7 | 0.92 | 0.87 | 0.82 | 0.74 | 0.74 | 0.66 | 0.66 | 0.67 | 0.62 | 0.66 | 0.66 | 0.63 | 0.61 | 0.60 | 0.59 |
| ε = 0.4 | 0.93 | 0.86 | 0.82 | 0.73 | 0.72 | 0.67 | 0.66 | 0.66 | 0.61 | 0.65 | 0.66 | 0.65 | 0.63 | 0.62 | 0.61 |

Figure 5: dCKA heatmaps for DenseNet-121

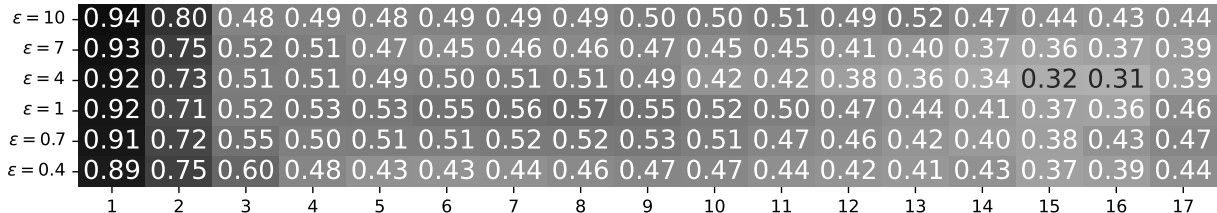

| ε | 1 | 2 | 3 | 4 | 5 | 6 | 7 | 8 | 9 | 10 | 11 | 12 | 13 | 14 | 15 | 16 | 17 |
|---|---|---|---|---|---|---|---|---|---|---|---|---|---|---|---|---|---|
| ε = 10 | 0.94 | 0.80 | 0.48 | 0.49 | 0.48 | 0.49 | 0.49 | 0.49 | 0.50 | 0.50 | 0.51 | 0.49 | 0.52 | 0.47 | 0.44 | 0.43 | 0.44 |
| ε = 7 | 0.93 | 0.75 | 0.52 | 0.51 | 0.47 | 0.45 | 0.46 | 0.46 | 0.47 | 0.45 | 0.45 | 0.41 | 0.40 | 0.37 | 0.36 | 0.37 | 0.39 |
| ε = 4 | 0.92 | 0.73 | 0.51 | 0.51 | 0.49 | 0.50 | 0.51 | 0.51 | 0.49 | 0.42 | 0.42 | 0.38 | 0.36 | 0.34 | 0.32 | 0.31 | 0.39 |
| ε = 1 | 0.92 | 0.71 | 0.52 | 0.53 | 0.53 | 0.55 | 0.56 | 0.57 | 0.55 | 0.52 | 0.50 | 0.47 | 0.44 | 0.41 | 0.37 | 0.36 | 0.46 |
| ε = 0.7 | 0.91 | 0.72 | 0.55 | 0.50 | 0.51 | 0.51 | 0.52 | 0.52 | 0.53 | 0.51 | 0.47 | 0.46 | 0.42 | 0.40 | 0.38 | 0.43 | 0.47 |
| ε = 0.4 | 0.89 | 0.75 | 0.60 | 0.48 | 0.43 | 0.43 | 0.44 | 0.46 | 0.47 | 0.47 | 0.44 | 0.42 | 0.41 | 0.43 | 0.37 | 0.39 | 0.44 |

Figure 6: dCKA heatmaps for EfficientNet-V2

mechanism to each pixel in each channel, scaling by $(k-1)nc/\epsilon$, such noise can obscure image semantics, particularly for smaller $\epsilon$. To mitigate this, Fan (Fan, 2018) advises dividing the image into $b \times b$ grids, averaging each grid's pixels, which reduces the sensitivity $(\Delta f)$ to $(k-1)nc/b^2$. Therefore, for images (including heatmaps) with a single channel, the sensitivity becomes $255n/b^2$. We empirically tested with a diverse set of $(n, b)$ for LDP explanations and communicated the results with the concerned physicians. We get the best response for the tuple $(16, 14)$, and the most competent explainers were `Grad-Shap` and `Integrated Gradients`. In our case, we get useful results $\epsilon = 4$ onwards. We selected the ResNet-34 to demonstrate our results for all four explainers, which can be found in the supplementary material.

### 8.1 What Could Be the Notion of PIS and LA Here?

After *LDP-fying* with $b \times b$ grids, pixel-wise comparisons are impractical. However, the postulates of LA do hold here as well: the most competent explanations should be least affected by LDP transformations. Therefore, such private explanations ($s'$) should be most *similar* to their non-private counterpart ($s$) and serve as a close proxy. To quantify the degradation of the *quality* of $s'$, we employ the Structural Similarity Index (SSIM) (Wang et al., 2004) as PIS in this context. In the absence of pixel-by-pixel comparison, unlike global DP, SSIM aims to reflect how much essential information is retained post-*LDP*fication, primarily focusing on several perceptual similarities by evaluating structural components in $s'$. However, as balancing the degree of noise to achieve a desired privacy level may severely affect the structural fidelity of $s'$, thus inevitably diminishes SSIM, we primarily use SSIM evaluation as an *elimination test* to discard any obfuscated *LDP*-fied explanations (having substantially low, near zero, or negative SSIM). Furthermore, we *eliminated* `Grad-CAM` as its small region of interests ($7 \times 7$ or $8 \times 8$) tend to get overwhelmed by *LDP* noise, and fixing the new set of hyperparameters here is also challenging. We roughly get PIS between $0.4 - 0.5$ for all other explainers

(Table 2); however, the concerned physician recommended `Integrated Gradients` and `Grad-Shap` to be *most useful* throughout our experiments.

|  | Grad-SHAP | Integrated Gradient | Saliency | SmoothGrad |
|---|---|---|---|---|
| Tuberculosis | 0.52 | 0.50 | 0.49 | 0.52 |
| Pneumonia | 0.53 | 0.51 | 0.43 | 0.49 |
| Normal | 0.56 | 0.54 | 0.49 | 0.53 |

Table 2: Mean SSIM for *LDP*-fied explanations.

In this setup, the model is non-private, but the explanations are private; we name this setup as Hybrid DP. Our novel software is outlined using this setup in Figure 7.

⊙ To summarize:

|  | Drop in Accuracy | Private Model | Private Output | Private Explanation | Useful Explanation |
|---|---|---|---|---|---|
| Global DP | ✓ | ✓ | ✓ | ✓ | ✗ |
| Hybrid DP | ✗ | ✗ | ✗ | ✓ | ✓ |

Table 3: Comparison of Privacy Methods

## 9    The Novel Software Pipeline

So far, we have investigated non-trivial intricacies of DP models and XAI methods, and we *figured out* an alternative way to generate private explanations using LDP. Based on all the insights we have consolidated so far, now we propose a privacy-preserving software pipeline that aims to reconcile model explainability and privacy[13].

**System Overview and Security Measures**: At the system's entry point, all incoming medical images pass through a trained autoencoder (AE) (Neloy and Turgeon, 2024), which filters out anomalous inputs. We achieve 94% mean accuracy on anomaly detection at a reconstruction loss threshold ($\kappa = 0.07$), using randomly sampled 50 test images from each non-target class (Cardiomegaly, Aortic enlargement from (Nguyen et al., 2022) dataset) as anomalous examples. Anomalous sample classification can be done with various methods, but we chose AE due to its widespread use in anomaly detection (Neloy and Turgeon, 2024).

**Core Processing Pipeline**: Validated images go through the core processing pipeline, where a non-private model makes predictions. The model's prediction (only the label) and the LDP-fied explanation (described in Section 8) are generated. These explanations are assessed for quality using SSIM (as shown in Figure 7). Explanations with poor SSIM scores (e.g., negative or near-zero values) are discarded to avoid wasting the physician's time.

The remaining top-K highest-quality LDP-fied explanations (here, K = 1-2), along with the model's prediction, are securely transmitted to the physician via an encrypted channel. This information is sent to the physician(s), not the patient, at this moment.

**Final Communication Protocol**: The final prediction and explanation, upon physicians' approval, are communicated to the patient.

■ How much 'Compromise' is required to 'Reconcile'? In our proposed software pipeline, we refute DP models for reconciling privacy and explainability in high-stakes, as XAI methods and DP models don't *go together*. Therefore, we must clearly *compromise* as we have to fall back to *non-private* model when we aim to gracefully *balance* both privacy and explainability in high-stakes. With LDP-fied explanations, we mitigate the privacy breach from explanation (as mentioned in Section 3) and we do not share the prediction vector, but

---

[13]This proposed Software Pipeline is *exclusively* based upon our exhaustive findings and insights that we have elaborated throughout our study, we do not claim this to be the (only) ideal setup for high-stakes.

only the label, to protect against MIA attacks from it. However, we acknowledge that our proposed pipeline is inappropriate for systems with a sheer requirement of both prediction vector and *useful* private explanations. From our findings, it is clear that, at least with DP models, building such systems is not possible. One may argue that if we LDP-fy the prediction vector coming from a non-private model by adding calibrated noise, it could be a *better* option than discarding the non-private prediction vector. We do not advocate making the prediction vector LDP by injecting noise, as excessive noise could alter the *actual* prediction and/or mislead decision-makers with false prediction confidence, impairing both model interpretability and utility. We note that sharing the model's prediction (only the label) is the least possible amount of *privacy risk* we take to make our pipeline *functional*. Our findings show that reconciling privacy and explainability in high-stakes situations comes with several other *silent factors* that need to be exclusively addressed, and is non-trivial. While this study aims to serve as a rigorous starting point, canonically reconciling privacy with explainability, with *minimum compromise*, is still an open challenge that demands further investigation, probably even beyond the setup and constraints we discussed in this paper.

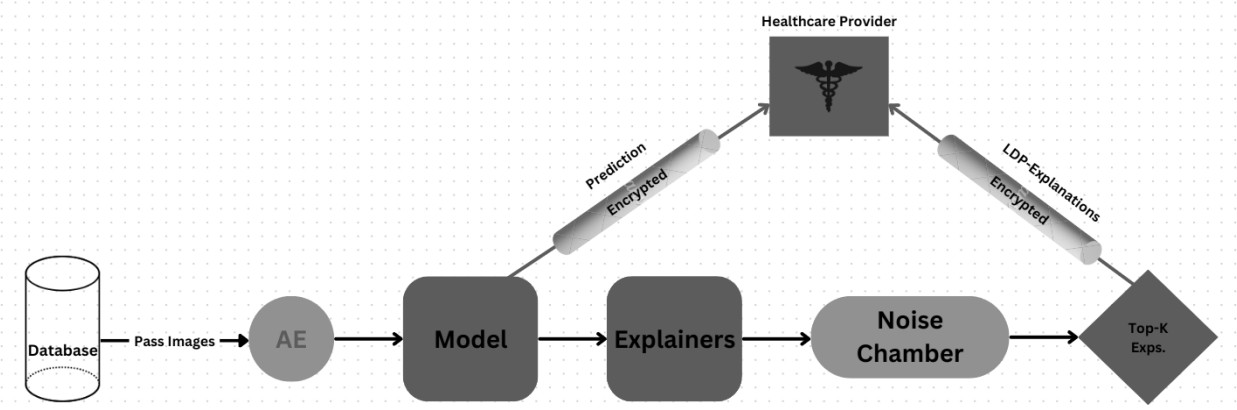

Figure 7: Outline of the Software

## 10 Related Work

*Privacy-preserving machine learning (PPML)* and *Explainable AI (XAI)* are well-studied research areas, and we've outlined the topics under consideration for this study in section 2. For a comprehensive review of *PPML* and *XAI*, we direct readers to the excellent surveys by Boulemtafes et al. (Boulemtafes et al., 2020) and Saeed et al. (Saeed and Omlin, 2023), respectively.

*Privacy-preserving explainable AI (PPXAI).* PPXAI methods are primarily an emerging class of XAI methods specifically made to protect *sensitive information* in their explanation (desirably with provable guarantees), primarily about the training data. PPXAI methods have only recently begun to emerge, with approaches such as differentially private Locally Linear Maps (LLM) (Harder et al., 2020), differentially private feature-based model explanations (Patel et al., 2022), and differentially private counterfactual explanations (Mochaourab et al., 2021; Yang et al., 2022), etc. Harder et al. in (Harder et al., 2020) proposed a family of simple models to *approximate* complex models using locally linear maps for each class, which aims to provide DP explanations. Patel et al. (Patel et al., 2022) introduced a perturbation-based algorithm that aims to make the *explanation dataset* DP, using which they, in a *black-box* fashion, generate a private explanation for a given *point-of-interest*. Mochaourab et al. (Mochaourab et al., 2021) made a DP Support Vector Machine and introduced methods for generating *robust* counterfactuals, Yang et al. (Yang et al., 2022) utilized a DP autoencoder that creates privacy-preserving prototypes by perturbing input data to minimize distance to a counterfactual, while promoting a specific class outcome. For a detailed overview of PPXAI, we direct readers to the comprehensive survey by Nguyen et al. (Nguyen et al., 2024).

Our work is completely orthogonal to PPXAI methods. It is worth noting that our work doesn't propose or analyze *any* PPXAI method but explains the interplay between off-the-shelf, well-regarded explainers

with DP models. We first establish the desiderata for DP explanations and then mechanistically explore why DP models are particularly challenging to explain using traditional post-hoc explainers. We not only investigate global DP and its inherent limitations for RTE but also explore an alternative route to generate *useful* private explanations using local DP to generate proxy private explanations.

Recently, researchers have begun investigating the quality of explanations in privacy-preserving environments. Bozorgpanah et al. (Bozorgpanah et al., 2022) generated private datasets (benchmark tabular datasets) using masking methods and compared the Shapley values (Lundberg and Lee, 2017) for test instances given the two models: trained on non-private data and trained on private data. They found minimal effect on moderate protection. Lucieri et al. (Lucieri et al., 2023) investigated the effect of DP training on concept-based explanations on biomedical datasets, and found that DP decreases average Concept Activation Vectors (CAV) accuracy and increases standard deviation; whereas (Saifullah et al., 2024) has reported that differential privacy and federated learning may yield 'noisy' feature attribution scores for post-hoc explainers. Berning et al. (Berning et al., 2024) found that k-anonymity degrades the quality of counterfactual explanations on a tabular dataset. Bozorgpanah et al. (Bozorgpanah and Torra, 2024) generated private datasets (benchmark tabular datasets) using masking methods and noise addition and applied TreeSHAP to achieve plausible explanations.

Our work shares some similarities with the aforementioned studies, but it differs significantly in approach. While previous research typically uses quality measures from the XAI literature in an ad-hoc manner within privacy-preserving environments, our systematic investigation is grounded in our own proposed postulate and measures: the Localization Assumption (LA) and Privacy Invariance Score (PIS). While earlier works focus on measuring the *degradation* in explanation quality within privacy-preserving settings using a limited set of explainers, we demonstrate under a tightly controlled environment that the issue isn't simply any *degradation* in explanation quality. Rather, it's the intrinsic nature of DP models themselves that leads to disobeying LA and producing orthogonal explanations across all commonly used gradient-based explainers, and we are the first ones to mechanistically interpret this very phenomenon. Moreover, many prior studies select explainers and quality measures in a rather ad-hoc fashion. In contrast, we acknowledge our apprehension regarding the suitability of model-agnostic explainers for DP models. Additionally, while Lucieri et al. (Lucieri et al., 2023) and Saifullah et al. (Saifullah et al., 2024) have focused on global DP, the majority of existing research has dealt with local DP.

*Analyzing network similarity.* Analyzing neural network similarity is crucial for interpreting and improving model behavior. This can be broadly classified into two main types: (i) representational similarity (RS), which measures differences in intermediate layer activations, and (ii) functional similarity, which evaluates discrepancies in model outputs (Klabunde et al., 2025).

To investigate why DP models fail to accommodate common post-hoc explainers, we investigated RS to examine layer-wise changes in activations and their sensitivities. However, in such cases, the similarity measures researchers employ adhere to specific invariances: Permutations, Orthogonal Transformations, Isotropic Scaling, Translations, etc, are to name a few. We have considered the most commonly used set of invariances and chose CKA for measuring RS, but CKA, by its design, comes with a few shortcomings as discussed in section 7. Consequently, we chose dCKA proposed by Cui et al. (Cui et al., 2022) for our analysis. For a broader view on neural network similarity, we refer the readers to the wonderful survey by Klabunde et al. (Klabunde et al., 2025).

## 11 Conclusion and Future Studies

We started our research with a simple question of whether we can achieve RTP and RTE *together*. We kept an eye on the pitfalls of commonly used evaluation metrics for explainability and proposed our desiderata, and found that no commonly used gradient-based explainers are useful for private models. We investigate the activations inside a DP model and how the model is sensitive towards those; we discovered that the intrinsic behaviour of DP models is the key reason behind this. Our *mechanistic* insights of the private models across different privacy guarantees highlight how DP training alters the internal representations and their sensitivities. It gives a fresh perspective on interpreting DP models and their behaviour from a nuanced

angle. Lastly, we make use of LDP to achieve private explanations and conclude our study by outlining the pipeline for the industrial software for our use case that respects both RTP and RTE.

In future studies, we aim to go deeper into the sensitivity landscape of DP models by investigating second-order and higher-order derivatives of the model's output w.r.t. representations, which may be helpful while investigating a few niche explainers' behavior (e.g.: Integrated Hessian (Janizek et al., 2021)), which do not directly leverage *immediate* first-order gradients. However, such explainers are still not widely used in high-stakes unlike the ones we have considered in this paper. While this work elucidates the interplay between the highly regarded gradient-based explainers and DP models, and *explains* the degradation of explanations of DP models, examining higher-order derivatives could potentially uncover richer structural patterns in the sensitivity space that are not immediately apparent through first-order analysis.

## Acknowledgement

We are thankful to Jim Conant for the fruitful discussion on high-dimensional geometry. We are thankful to Yogesh Kumar, the co-author of dCKA (Cui et al., 2022), for a productive discussion. We are also thankful to the physician from SRM University AP, Dr. Venkata Abhinay Talasila, for his immense cooperation throughout.

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

## Appendices

### A    CIFAR-10 Dataset Reslts

We trained ResNet-34 and DenseNet-121 models on the CIFAR-10 dataset using three $\epsilon$ values $(4, 7, 10)$, as lower $\epsilon$ values resulted in harsh privacy-utility trade-off. We weren't able to train EfficientNet-v2 due to its exuberant computational requirements. We evaluated the models on the whole test set. We train all models (both DP and non-DP counterparts) with the exact set of hyperparameters over 50 epochs. For brevity, we report $Acc_{\mathcal{M}}$ (Table 4), $Acc_{\mathcal{M}'/\mathcal{M}}$, and $-\times-$ taking all classes together for *Perf Comp.* and *Agreement* in Figure 10.

However, in this case as well `Integrated Gradients` and `Grad-Shap` yield 30% as mean DS score, and the rest of the explainers also do not obtain $PIS_{Avg} > 0.3$ (Figure 10). Furthermore, here also we do not find any (apparent) $PIS_{Avg}$ follows with *Perf Comp.* and/or *Agreement.* All in all, the results from our primary experiment are sufficiently comparable here as well. We will release the weights of these models upon publication.

For ResNet-34, we observed that sensitivity is independent across all layers for $\epsilon = 10$. However, only the last 2-3 layers for other $\epsilon$ values for ResNet-34 exhibited independent sensitivity. It indicates that independent sensitivity, even in the last few layers, can potentially make the explanations incomparable across $\epsilon$ values. In contrast, for DenseNet-121, independent sensitivity was consistently observed for all layers across all $\epsilon$. For both types of models, we obtained fewer than 5–7% of layers where we couldn't reject the null hypothesis for independence of representation. We have reported the dCKA heatmap for DenseNet-121 and ResNet-34 models in Figure 9, 8, respectively. Notably, the final layer for both ResNet-34 and DesneNet-121 consistently demonstrated independent sensitivity, rendering `Grad-CAM` unsuitable for generating comparable explanations. Consequently, we focused on other explainability methods in Figure 10.

| Model | Acc (%) |
|---|---|
| DenseNet-121 | 72.82 |
| ResNet-34 | 66.61 |

Table 4: Non-private models' accuracy

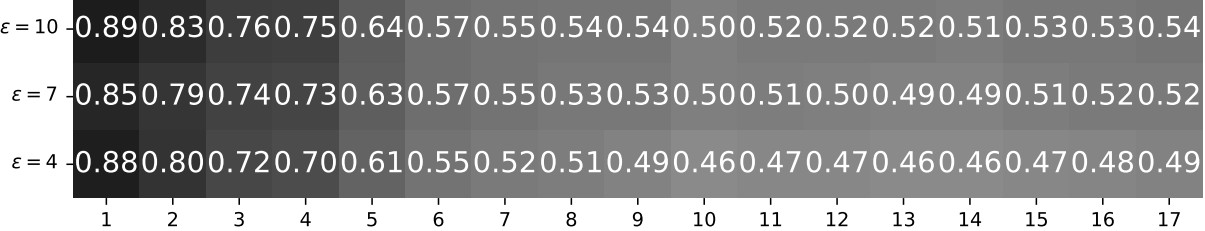

Figure 8: dCKA heatmap for ResNet-34 for CIFAR-10.

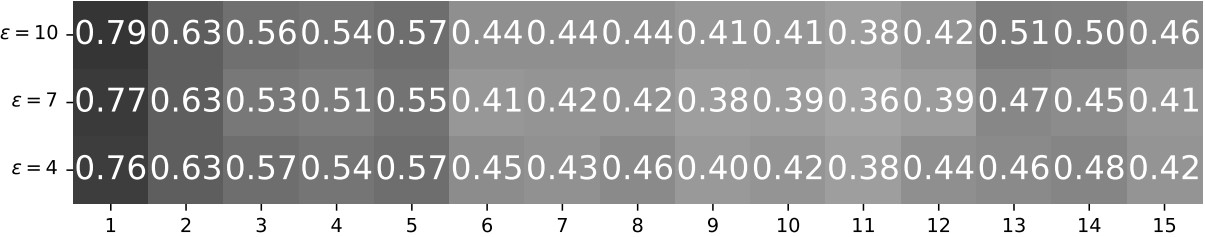

Figure 9: dCKA heatmap for DenseNet-121 for CIFAR-10.

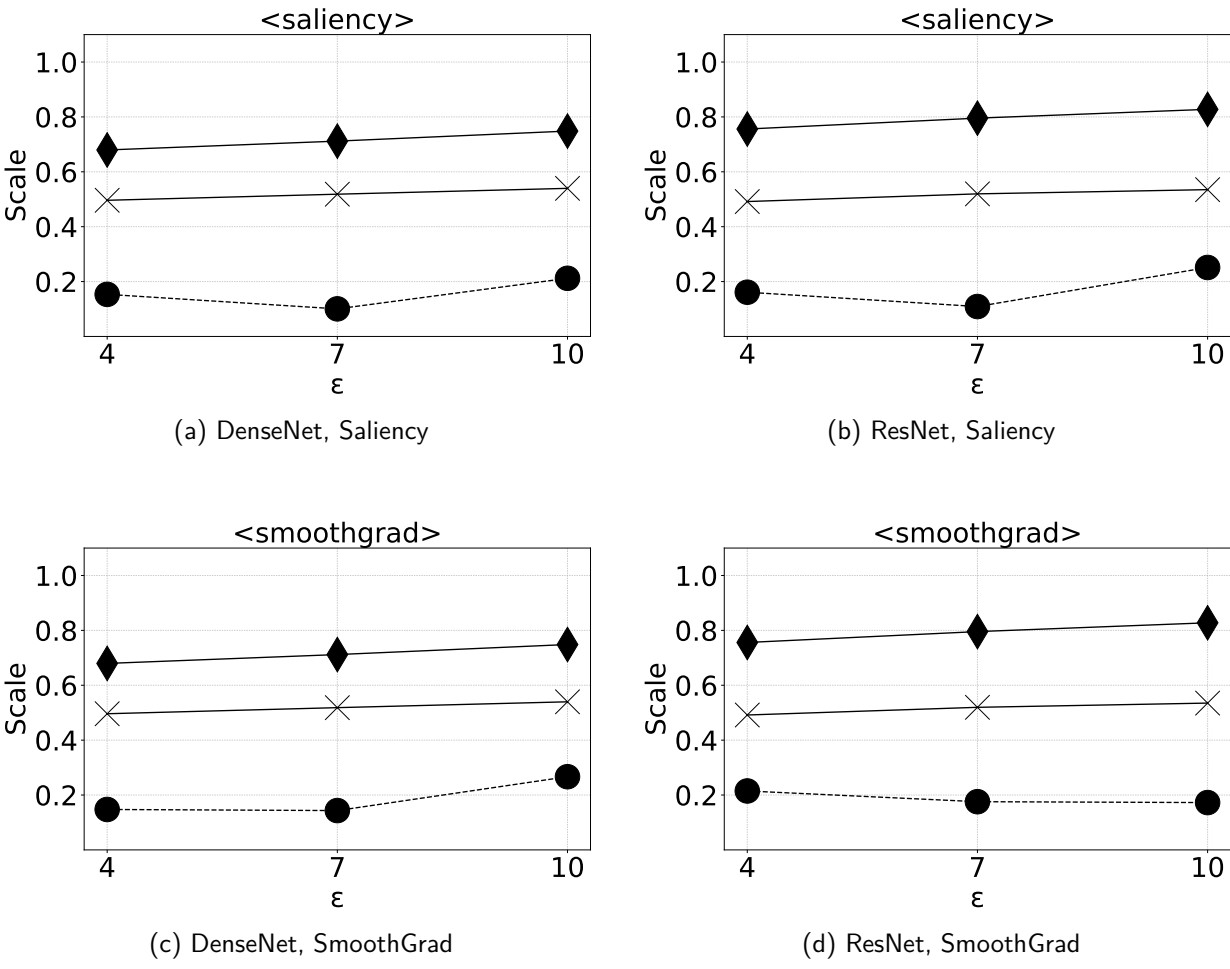

Figure 10: Analysis on CIFAR-10: - ● - for $PIS_{Avg}$, $-\blacklozenge-$ for $Acc_{\mathcal{M'}/\mathcal{M}}$, and $-\times-$ for *Agreement* between non-private and private model pair.

## B    Details of the Explainers

**LIME (Local Interpretable Model–Agnostic Explanations):**   LIME (Ribeiro et al., 2016) approximates a complex model locally using a simpler interpretable model (e.g., linear regression). For an input $x$, LIME generates perturbed versions of $x$ and calculates the corresponding predictions to fit a local surrogate model.

Mathematically, LIME solves the following optimization problem:

$$\arg\min_{g \in G} \mathcal{L}(f, g, \pi_x) + \Omega(g),$$

where $f$ is the original model, $g$ is the interpretable surrogate model, $\pi_x$ is a proximity measure to $x$, and $\Omega(g)$ ensures simplicity of $g$.

**SHAP (SHapley Additive exPlanations):** SHAP (Lundberg and Lee, 2017) explains model predictions based on cooperative game theory. For a prediction $f(x)$, SHAP attributes contributions to each feature using Shapley values:

$$\phi_i = \sum_{S \subseteq N \setminus \{i\}} \frac{|S|!(|N| - |S| - 1)!}{|N|!} \left[ f(S \cup \{i\}) - f(S) \right],$$

where $N$ is the set of all features, $S$ is a subset of features, and $f(S)$ is the model's prediction with features in $S$ included.

**Saliency Maps:** Saliency maps (Simonyan et al., 2013) visualize the importance of each input feature by computing the gradient of the output $f(x)$ with respect to the input $x$. Standard implementations (by default) use absolute values of the gradients (Kokhlikyan et al., 2020).

$$\text{Saliency}(x_i) = \left| \frac{\partial f(x)}{\partial x_i} \right|$$

**SmoothGrad:** SmoothGrad (Smilkov et al., 2017) reduces noise in saliency maps by averaging gradients over multiple noisy samples of the input:

$$\text{SmoothGrad}(x) = \frac{1}{n} \sum_{i=1}^{n} \nabla_x f(x + \mathcal{N}(0, \sigma^2)).$$

**Integrated Gradients:** Integrated Gradients (Sundararajan et al., 2017) attribute feature importance by integrating the gradients along the path from a baseline input $x'$ to the actual input $x$:

$$\text{IG}_i(x) = (x_i - x_i') \int_{\alpha=0}^{1} \frac{\partial f(x' + \alpha(x - x'))}{\partial x_i} d\alpha.$$

**Grad-CAM:** Grad-CAM (Selvaraju et al., 2019) generates heatmaps for convolutional neural networks by using gradients of the target output with respect to feature maps of a convolutional layer. For a given feature map $A_k$, the weights are computed as:

$$\alpha_k^c = \frac{1}{Z} \sum_i \sum_j \frac{\partial y^c}{\partial A_k^{ij}},$$

where $y^c$ is the output score for class $c$, and $Z$ is the spatial dimensions of $A_k$. The Grad-CAM heatmap is:

$$\text{Grad-CAM} = \text{ReLU} \left( \sum_k \alpha_k^c A_k \right).$$

## C    Implimentation Details

### C.1    Additional Details on Experimental Setup

We train the non-private and private models fixing all the hyperparameters (batch size: 128, lr: 0.001, delta (for DP): 0.001) except for the number of epochs, as private models need more computation to learn due to the heavy regularization DP introduces in the training (Ponomareva et al., 2023). Following (Ponomareva et al., 2023), we initialised all our models (both non-private and private counterparts) with publicly available pre-trained weights (ImageNet) for *better* convergence. Furthermore, to make a fair comparison we have fixed the number of all hyperparameters in all the private models with different $\epsilon$. We set the no. of epochs as 50 for all private models; it yielded competitive accuracy across model types. However, within $13 - 16$ epochs, all non-private models achieved accuracy $> 95\%$.

However, while training, we discovered an issue in the vanilla architecture of the aforementioned networks: they utilize batch normalization (`BatchNorm`). Nevertheless, `BatchNorm` normalizes a sample based on the statistics of the batch it is in. This means the same sample can get different normalized values depending on the other samples in the batch. For differential privacy, each sample's privacy needs to be independently preserved. Since `BatchNorm` depends on other samples, it violates this principle and leaks information about other samples. (Yousefpour et al., 2021) advises replacing `BatchNorm` layers with privacy-friendly options like Group Normalization, Layer Normalization, Instance Normalization, etc. From the engineering perspective, we have to select one such replacement that scales with sufficiently large datasets without hampering the privacy bounds. Based on previous empirical evidence (Subramani et al., 2021), we replace the `BatchNorm` layers with `GroupNorm` layers in all non-private models along with their private counterparts, as `GroupNorm` does not alter the base architecture drastically, scales well, and adheres to the privacy principle strictly. We utilized the `Opacus` library for DP-training (`https://opacus.ai`).

We employ the off-the-shelf, publicly available implementations of the explainers from `Captum` library (Kokhlikyan et al., 2020).

### C.2    Implementation Details for Section 7

For (d)CKA, we utilized the publicly available package: `Simtorch` (`https://github.com/ykumards/simtorch`) with default (hyper)parameter selection. For Statistical testing with HSIC, we utilized the publicly available package: `PyRKHSstats` (`https://github.com/Black-Swan-ICL/PyRKHSstats`) with default (hyper)parameter selection except for the default p-value cutoff of 0.01. We have used a p-value threshold of 0.05 throughout our experiments. In all our experiments, we considered the whole test set at once as a single batch for dCKA calculation and statistical testing with HSIC.

We run all our experiments on an NVIDIA DGX workstation, leveraging 1 Tesla V100 32GB GPU. We wrote all experiments in Python 3.10. Our total computational time for all experiments is roughly 81 hours.

