## Supplementary Material

We selected the ResNet-34 for demonstrating our results and we have reported explanations from all four explainer in Figure 1 and 2.

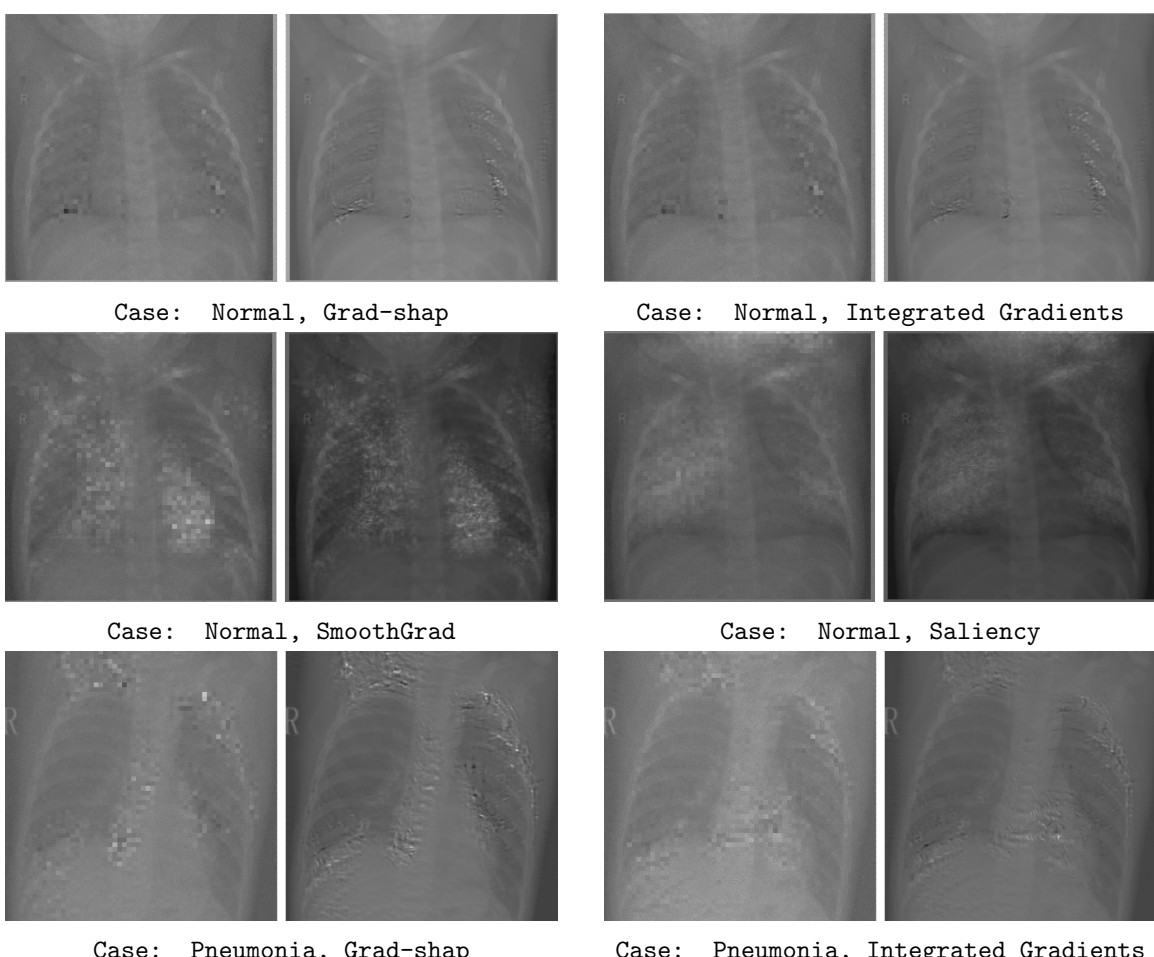

Figure 1: For each plot, LDP explanation is on **Left** and Vanilla explanation is on **Right**

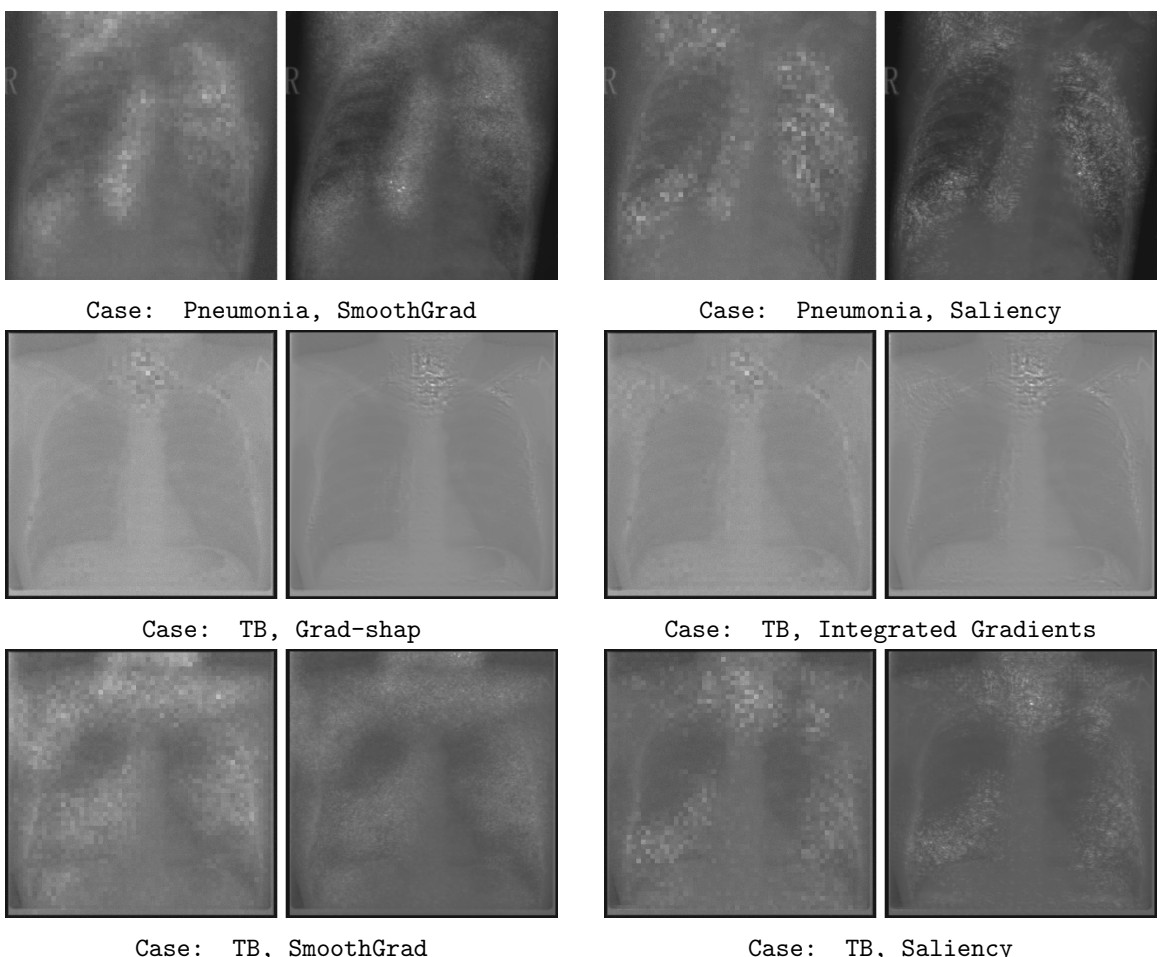

Figure 2: For each plot, LDP explanation is on **Left** and Vanilla explanation is on **Right**