# OpenReview forum: "Reconciling Privacy and Explainability in High-Stakes: A Systematic Inquiry"
_TMLR — Accepted by TMLR_

### Review · Reviewer_SovN · 2025-02-28

**Summary Of Contributions:**

This paper addresses the challenge of reconciling privacy preservation via DP and explainability in high-stakes AI domains. The study makes three primary contributions. First, they systematically analyze the compatibility of gradient-based post-hoc explainers with DP-enhanced models, revealing that DP-induced shifts in model representations and sensitivities would possibly destroy explanation quality. This work highlights a notable gap in existing research that the interplay between privacy-preservation and explainability accuracy remains poorly understood. Second, the authors propose the Localization Assumption (LA) and Privacy Invariance Score (PIS), a novel evaluation measurement to quantitatively assess alignment between explanations from DP and non-DP models. These tools provide structured metrics to assess explainability in privacy-sensitive contexts and address the limitations of previous faithfulness measures. Third, they introduce a hybrid DP pipeline that combines non-DP models with locally differentially private (LDP) explainability, balancing model performance and privacy preserving by privatizing explanations through noise injection. This approach offers a practical solution for deploying explainable AI in regulated domains like healthcare, ensuring compliance with both Right-to-Privacy (RTP) and Right-to-Explanation (RTE).

**Audience:**

Yes

**Broader Impact Concerns:**

Not applicable.

**Claims And Evidence:**

Yes

**Requested Changes:**

1. Expand threat models to include adaptive adversaries using explanations for adversarial examples or MIA. Analyze multi-stage attacks exploiting explanations to uncover reasoning and launch downstream attacks. For instance, combining DP-enhanced outputs with non-private explanations and auxiliary data could reconstruct sensitive inputs. Comprehensive models are needed to address direct and indirect risks in privacy-sensitive systems.
2. Analyze sensitivity landscapes to identify robustness patterns via higher-order derivatives. Test SHAP, LIME, and Grad-CAM under adaptive attacks. Propose countermeasures like LDP with gradient masking. Provide empirical evidence for DP-SGD and LDP’s effectiveness in mitigating attacks, ensuring secure explanations under adversarial conditions.
3. Propose a framework to evaluate explanation faithfulness in privacy-preserving settings, combining qualitative and quantitative measures. Assess existing metrics like Infidelity for shortcomings and introduce alternatives. Integrate expert oversight for high-stakes standards. Explore alignment-based metrics for contexts where traditional faithfulness is unreliable, ensuring private explanations remain meaningful.

**Strengths And Weaknesses:**

Strengths

1. The paper provides a rigorous privacy-explainability analysis which evaluates several gradient-based explainers (SmoothGrad, Grad-Shap, Grad-CAM, etc.) across three CNN architectures (ResNet-34, DenseNet-121, EfficientNet-V2) and multiple privacy budgets. For instance, Grad-CAM explanations degrade significantly under DP, with PISAvg scores never exceeding 0.32 (Section 6, Figures 1–3). This low score indicates that DP models produce explanations that poorly align with non-private counterparts, even when predictions match. The study reveals that DP-induced noise disrupts gradient-based attribution, making traditional explainers unreliable for auditing private models.
2. The work proposes a novel evaluation framework. The Localization Assumption (LA) enforces a 15% disagreement threshold for attribution type mismatches, ensuring explanations from DP and non-private models align structurally. The Privacy Invariance Score (PIS) uses Kendall Tau correlation to measure similarity between explanations (Section 4.3). For example, LA filters out Grad-CAM explanations with >15% disagreement, while PIS quantifies the remaining explanations’ alignment. This framework addresses pitfalls of traditional metrics like faithfulness, which often fail in DP settings due to altered model sensitivities.
3. The work offers a comprehensive experimental study. The experiments span two datasets (Chest X-ray, CIFAR-10) and include statistical significance via HSIC (p<0.05). For example, LDP explanations on CIFAR-10 show PISAvg < 0.3 for all explainers (Appendix A, Figure 10), replicating the primary findings. The study also tests scalability limits, noting EfficientNet-V2’s computational demands under DP (Section 5.1).

Weakness:

1. Incomplete threat models in PPML and XAI leave vulnerabilities unaddressed. For instance, DP protects training data, but post-hoc explainers like SHAP or LIME can leak sensitive information, enabling membership inference. In medical applications, Grad-CAM heatmaps could expose patient data. Many models also overlook multi-stage attacks, where explanations reveal reasoning, enabling adversarial examples or MIA. Combining DP-enhanced outputs with non-private explanations and auxiliary knowledge could reconstruct sensitive inputs. Comprehensive threat models are needed to address these risks.
2. PPML and XAI techniques often lack robustness against adaptive attacks, where adversaries dynamically adjust strategies. For instance, DP’s noise can be reverse-engineered, and gradient-based explainers can be manipulated via perturbations, producing misleading explanations. This is critical in high-stakes applications. DP noise also disrupts explainability, creating exploitable inconsistencies. Solutions must balance robustness, utility, and privacy.
3. Theoretical gaps exist in integrating DP with explainers, as extending DP’s privacy guarantees to explanations lacks a solid foundation. Adding noise (e.g., via LDP) ensures privacy but lacks proof of preserving faithfulness or utility. Without rigorous bounds, compliance with regulations like RTE is uncertain. Theoretical analyses often assume ideal conditions, invalidating real-world claims. Evaluation metrics like Infidelity often lack validity, complicating quality assessment. A unified framework is needed to reconcile privacy and explainability.

---

> ### Author Response · Authors · 2025-04-05
> **Authors's Response (Part 1/n)**
>
> We wholeheartedly appreciate your thorough and prompt analysis of our paper in such a short time. We’d like to address your concerns as follows:
>
> W1+RC1
>
> ``Incomplete threat models in PPML and XAI leave vulnerabilities unaddressed. For instance, DP protects training data, but post-hoc explainers like SHAP or LIME can leak sensitive information, enabling membership inference. In medical applications, Grad-CAM heatmaps could expose patient data. Many models also overlook multi-stage attacks, where explanations reveal reasoning, enabling adversarial examples or MIA. Combining DP-enhanced outputs with non-private explanations and auxiliary knowledge could reconstruct sensitive inputs. Comprehensive threat models are needed to address these risks.``
>
> Our presented threat model is complete (section 3.1) and does take care of privacy breaches from explainers (second point), along with breaches from the output vector (point 1). DP comes with post-processing property (we have added one more reference for the convenience of the readers). When a mechanism (any explainer here) is applied to the DP model, the obtained explanations are also DP wrt the training data. Essentially, when we use a DP model, both the output and explanations are private due to post-processing, mitigating multi-stage explanation-based attacks, as you mentioned. However, in our study, we found that we cannot use DP models for generating useful private explanations due to DP’s intrinsic nature (section 7), and therefore, we had to take an alternative approach, leveraging LDP to make explanations private, which also prevents privacy attacks from explanations. Therefore, any multistage and/or adaptive privacy attacks at any stage, exploiting explanations, are prevented for both (global) DP models and the hybrid DP approach we proposed to reconcile privacy and explainability in high-stakes.
>
> However, we notice that you also mentioned adversarial examples. We note that adversarial attacks against both explainers [2] and models [1] are completely out of scope with our main research goal. However, we’d like to highlight that when we were designing our software pipeline (section 9), such requirements were taken care of (strictly from a system requirement, not from research setup’s perspective), and to prevent several types of anomalous inputs including adversarial examples from entering the pipeline, we placed a gatekeeper (in our case, a trained autoencoder which has been used extensively in prior study for preventing against anomalous inputs). Adversarial attacks against explainers are primarily perturbations in the input space that aim to yield very different explanations, even when the output from the model is the same (or similar) for both the actual and perturbed input. Even if our gatekeeper, for the sake of discussion, didn’t catch these abnormal inputs, as our final inspection is always contingent upon the decision maker’s approval. Thus, at least on the client’s side, such attacks won’t hamper them eventually. However, we use LDP-fied private explanations, and from that, investigating how much adversarial attacks on explanations are feasible is out of scope for our paper.

---

> ### Author Response · Authors · 2025-04-05
> **Authors's Response (Part 2/n)**
>
> WC-2
>
> ``PPML and XAI techniques often lack robustness against adaptive attacks, where adversaries dynamically adjust strategies. For instance, DP’s noise can be reverse-engineered, and gradient-based explainers can be manipulated via perturbations, producing misleading explanations. This is critical in high-stakes applications. DP noise also disrupts explainability, creating exploitable inconsistencies. Solutions must balance robustness, utility, and privacy.``
>
> Accessing PPML’s or XAI’s robustness does not come under our primary research scope. Our formal research questions and contributions are clearly stated in the introduction section. We have already addressed that adversarial attacks against explainers (“gradient-based explainers can be manipulated via perturbations, producing misleading explanations.”) don’t come under our study’s scope, and please refer to our reply on wc1+rc1. We are not aware of reverse engineering the DP noise, but we think that even if we consider such constructs, that will be completely orthogonal to our research goal, as once a mechanism is DP-fied, its immunity against MIA is independent of the knowledge of an adversary regarding the noise used to make it DP.
>
> ``DP noise also disrupts explainability….`` – Our central research question revolves around a similar quest: Do DP models and regular post-hoc explainers go together? We have craft fully answered this. To give an outline of our work, once again, we quote our conclusion section:
>
> “We started our research with a simple question of whether we can achieve RTP and RTE together. We kept an on eye the pitfalls of commonly used evaluation metrics for explainability and proposed our desiderata and found that no commonly used gradient-based explainers are useful for private models. We investigate the activations inside a DP model and how the model is sensitive towards those; we discovered that the intrinsic behaviour of DP models is the key reason behind this. Our mechanistic insights into the private models across different privacy guarantees highlight how DP training alters the internal representations and their sensitivities. It gives a fresh perspective on interpreting DP models and their behaviour from a nuanced angle. Lastly, we make use of LDP to achieve private explanations and conclude our study by outlining the pipeline for the industrial software for our use case that respects both RTP and RTE.”
>
> ``Solutions must balance robustness, utility, and privacy.``
>
> Our postulates (LA) check both the utility of the explanations in the DP environment and also take care of the utility for DP models (Perf Comp. and Alignment in section 4.2). Furthermore, we also do extensive empirical experiments to set up the hyperparameters for LDP-fied explanations and validate with physicians to preserve both their privacy and subjective utility in Hybrid DP. Our research is primarily not intended to be a watertight engineering solution for high-stakes taking care of everything. However, from our findings, we presented a software pipeline that addresses the implicit need for robustness against anomalous inputs. Hence, our solution balances robustness, utility and privacy.
>
> RC - 2
> ``Analyze sensitivity landscapes to identify robustness patterns via higher-order derivatives. ``
>
> The popular off-the-shelf explainers, which we’ve used, utilize first-order gradients. We mechanistically examined first-order sensitivity to show why they are inappropriate for DP models (section 7). This is why we didn’t need to go beyond the first-order analysis of the sensitivity of the representation space layerwise in DP models and their non-DP counterparts. However, for a few other niche explainers, which are not limited to the first-order gradient, we may need to go beyond the point we have already explored. We had already mentioned the potential usage of higher-order derivatives in the future study section, and in this version, we have elaborated on the same for better understanding.
>
> ``Test SHAP, LIME, and Grad-CAM under adaptive attacks.``
>
> We’ve already clarified your concerns about adaptive attacks above.
>
> ``Propose countermeasures like LDP with gradient masking.``
>
> We already use LDP for an alternative approach to generate useful private explanations. We do not understand where else a countermeasure is required, and what is the usefulness of gradient masking + LDP there. If you still feel there is a sheer need for this, please justify the same, subject to our research questions and setup with necessary and sufficient references.
>
> ``Provide empirical evidence for DP-SGD and LDP’s effectiveness in mitigating attacks, ensuring secure explanations under adversarial conditions.``
>
> DP-SGD is the algorithm we use for DP training. Both DP-SGD and LDP are already well-known to have strong, quantifiable defenses. Therefore, we do not feel any additional need for empirical justification, at least for this study. Requested to refer to Section 5.2 for more details.

---

> ### Author Response · Authors · 2025-04-05
> **Authors's Response (Part 3/n)**
>
> W3:
>
> ``Theoretical gaps exist in integrating DP with explainers, as extending DP’s privacy guarantees to explanations lacks a solid foundation.``
>
> No “theoretical gap” exists. As stated before, post-processing ensures this. We’ve already mentioned this in section 3.1.
>
> Adding noise (e.g., via LDP) ensures privacy but lacks proof of preserving faithfulness or utility. Without rigorous bounds, compliance with regulations like RTE is uncertain. Theoretical analyses often assume ideal conditions, invalidating real-world claims. Evaluation metrics like Infidelity often lack validity, complicating quality assessment. A unified framework is needed to reconcile privacy and explainability.
>
> We appreciate your take on the unified framework to reconcile privacy and explainability. We envision that utility for (LDP-fied) explanation is a subjective context, and it has to align with the experts’ recommendations (section 3.4; remark 1, 2). Therefore, we have extensively worked with the hyperparameter selection and chose the one that is best recommended according to the concerned physician. Till this point, we have also clarified how we aim to reconcile privacy and utility with the help of the insights we gained through our extensive findings. For a better overview of our proposed unified framework for reconciliation, we have rewritten section 9.

---

> ### Author Response · Authors · 2025-04-05
> **Authors's Response (Part 4/n)**
>
> RC - 3
>
> ``Propose a framework to evaluate explanation faithfulness in privacy-preserving settings, combining qualitative and quantitative measures. Assess existing metrics like Infidelity for shortcomings and introduce alternatives.``
>
> Our study is not about faithfulness in privacy-preserving environments. In fact, we criticize the existing faithfulness measures, but we didn’t propose our own faithfulness evaluation measure. Firstly, we promptly consolidated the flaws in the commonly used faithfulness metrics to strengthen our motivation, and then we proposed two remarks that clarify our stand. In remark 2, we have stated that:
> “Explanations should align with local constraints and contexts, even when (so-called) faithfulness cannot be measured reliably.”
>
> In our context, we aim to use a proxy model (DP model) in place of another model (non-DP model), so we’ve introduced LA (section 5). It is self-explanatory that LA and remark 2 are orthogonal to faithfulness, and we primarily aim to measure the similarity (not faithfulness) b/w the explanations coming from DP and non-DP models for inputs where prediction matches. As PIS, with which we are quantifying LA, in essence, is a similarity metric, we cannot compare the same with any faithfulness metric. We note that the hypothesis of LA is inspired by a well-regarded postulate (the model assumption) in contemporary XAI literature, but LA, in itself, is not a faithfulness measure.
>
> However, we would like to emphasize that LA is pretty effective for the job it was designed for. It did show us that the explanations from DP models are substantially incomparable with their non-DP counterparts, which, in turn, has triggered our inquisition to investigate the internals of DP models, and we later figured out (in section 7) that: DP training was primarily meant to resist MIA w.r.t. the training data but the training goes much beyond the scope, and drastically alters the overall representation and sensitivity space of a model. Which, as demonstrated, makes the popular post-hoc methods nonfunctional.
>
> ``Integrate expert oversight for high-stakes standards.``
>
> We do incorporate expert (in our case, concerned physician) oversight to preserve the utility wherever required. In fact, we start our study by arguing regarding the same in section 3.4; remark 1. For better clarification, we refer to sections 8.1, 9.
>
> ``Explore alignment-based metrics for contexts where traditional faithfulness is unreliable, ensuring private explanations remain meaningful.``
>
> We do not understand what the alignment-based methods you’re referring to are. We assume that this is about the “plausibility” of the explanation. If that is the case, we already use SSIM as an elimination criterion to discard obfuscated LDP-fied explanations (section 8.1), and then we let the concerned subject-matter expert (here, the concerned physician) take the final decision (section 9). We do not feel any further need to incorporate any further measures that aim to quantify plausibility or alignment. It is well-known that different explainers may produce different explanations for the same test samples. Plausibility or alignment metrics may assign higher scores to an explanation that is factually flawed. In fact, plausible yet not faithful explanations are a common trap in several contexts [3]. This unnecessarily complicates the whole pipeline.
>
>
> 1. Chakraborty, A., Alam, M., Dey, V., Chattopadhyay, A., & Mukhopadhyay, D. (2018). Adversarial attacks and defences: A survey. arXiv preprint arXiv:1810.00069.
>
> 2. Baniecki, H., & Biecek, P. (2024). Adversarial attacks and defenses in explainable artificial intelligence: A survey. Information Fusion, 102303.
>
> 3. Zhong, R., Shao, S., & McKeown, K. (2019). Fine-grained sentiment analysis with faithful attention. arXiv preprint arXiv:1908.06870.

---

### Review · Reviewer_mvhp · 2025-03-25

**Summary Of Contributions:**

The paper investigates whether it is possible to have differential private models and private explanations at the same time. To investigate this, the paper explores post-hoc explanation methods such as saliency maps, grad-cam or smooth grad. As a use case disease detection on chest x-ray images is used. The study finds that post-processing gradient-based explainers perform poorly on DP models because the internal representations of a DP model is not similar to a non-DP model.
To provide private explanations, the paper proposes to use the non-private model for explanation generation and then apply Laplacian noise to the explanations.

**Audience:**

Yes

**Broader Impact Concerns:**

In my opinion no broader impact statement is required.

**Claims And Evidence:**

Yes

**Requested Changes:**

1) Please either show all the results of all models/explainers or clarify how and why the thresholds are calculated and why these results are not shown. Even when the explainers do not meet a certain threshold, it is still viable to show the results and explain why they perform poorly.
2) The main question of the paper is whether private model explanations can be given using DP models. However, this question is not entirely answered in my opinion. Instead, it is proposed to keep using the non-DP model and modify the explanation to be private. However, this defeats the purpose of using DP models in the first place, when trying to protect against MIAs and privacy attacks. As a result, I think either the research question needs to be adjusted to something like "Can DP models be used to generate private gradient-based explanations?" or another solution has to be proposed, being able to only use the DP model.
3) As far as I can understand, no formal guarantees for the privacy of the LDP-explanation can be given. This should be clearly mentioned and discussed.
4) Please add the paper of Shokri et al. [1] when introducing MIAs as this was the first paper that introduced this attack.
5) Please go through the paper once more and fix all the typos and improve the transitions of the sentences.

While 1), 2) and 3) are critical for acceptance, in my opinion, 4) and 5) would further improve the paper.

[1] Shokri et al., Membership Inference Attacks Against Machine Learning Models, IEEE S&P 2017

**Strengths And Weaknesses:**

Strengths:
- Most of the time, the paper reads well and is clearly written
- It is shown on two datasets and three different models that gradient-based explanations are bad on DP models

Weaknesses:
- The paper of Shokri et al. regarding membership inference attacks from 2016 is missing as a reference
- The transitions between the sentences in the overview of the paper at the end of section 1 could be improved.
- The hypothesis that explainers perform poorly on DP models is only tested on 2 datasets (X-Ray and CIFAR-10)
- For these experiments, showing that the explainers perform poorly on DP models, it is not easy to comprehend why some methods are not shown (e.g., because the threshold on the disagreement score is not met).
- In the first paragraph of section 6 it is written: "Across models, around 30 − 40% Grad-Cam explanations of test samples violated the DS threshold, we have considered the rest of the samples to calculate PIS."
  It is unclear to me why the samples have to be filtered using the DS threshold. If the two models do not agree, then the DP-model seems to be bad. Assuming that the user/provider of the model can filter data points is very unrealistic in my opinion as only the DP model is available.
- In my opinion it doesn't make sense to use the non-private model for predicting and using it for the explanations as the model is then still susceptible to MIAs. This should be clearly discussed and made clearer in my opinion.

Misc:
- Bottom of page 3: "Broadly, these explainers can be (broadly)" -> two times "broadly"
- Section 7: "righrous" -> rigorous
- Section 7 towards the end: "we were unable to reject the null hypothesis H_0 for accross..." -> either remove "for" or "across"
- Section 8: "Where A satisfies $\epsilon$-differential privacy" -> unnecessary line break and "Where" needs to be lower-cased
- Section 10 paragraph on "Analyzing network similarity": "Analzsing" -> typo, should be Analyzing

---

> ### Author Response · Authors · 2025-04-05
> **Authors's Response (Part 1/n)**
>
> We wholeheartedly appreciate your thorough analysis of our paper.  We’d like to address your concerns as follows:
>
> W1+RC4:
>
> ``The paper of Shokri et al. regarding membership inference attacks from 2016 is missing as a reference``
>
>
> ``Please add the paper of Shokri et al. [1] when introducing MIAs as this was the first paper that introduced this attack.``
>
> We have updated our manuscript with this citation. We acknowledge the extensive line of work Dr. Shokri and his coauthors have contributed to the field of PPML.
>
> W2:
>
> ``The transitions between the sentences in the overview of the paper at the end of section 1 could be improved.``
>
> Thank you for your suggestion! In the revised version of the paper, we have taken care of this.
>
> W4+W5+RC1:
>
> ``For these experiments, showing that the explainers perform poorly on DP models, it is not easy to comprehend why some methods are not shown (e.g., because the threshold on the disagreement score is not met).``
>
> In the first paragraph of section 6 it is written: "Across models, around 30 − 40% Grad-Cam explanations of test samples violated the DS threshold, we have considered the rest of the samples to calculate PIS." It is unclear to me why the samples have to be filtered using the DS threshold. If the two models do not agree, then the DP-model seems to be bad. Assuming that the user/provider of the model can filter data points is very unrealistic in my opinion as only the DP model is available.
> Please either show all the results of all models/explainers or clarify how and why the thresholds are calculated and why these results are not shown. Even when the explainers do not meet a certain threshold, it is still viable to show the results and explain why they perform poorly.
>
> We have rewritten sections 4.3 and section 6’s first paragraph to enhance the readability, addressing your points. DS for a pair of explanations (one is for M, another is for M’, where M(x) = M’(x), given an explainer) is defined as the % of pixels and/or elements where their corresponding attributes’ nature (positive, negative) differ. We excluded IG and Grad-Shap, as for each test sample, we found the disagreement score is at least 45% for any pair of models (M, M’). This means that almost half of the explanation space, when comparing, is mismatched. Although we set a threshold of 15% empirically for this mismatch in our case study, IG and SHAP produce substantially different explanations, which is why we remove them from further analysis. However, this doesn’t hamper our analysis, as we discovered stronger results while mechanistically interpreting the model. We found out that both the representation deteriorates and the sensitivity space layerwise are not comparable, due to which, all gradient-based explainers will produce orthogonal explanations along with IG and Grad-Shap.
>
> Regarding Grad-CAM, 30 – 40% of samples didn’t meet the DS threshold of 15%, so we discarded those samples. We did a quick ablation by varying the threshold (15, 20, 25%), but that doesn’t alter the trend. The primary reason behind that is that throughout all our experiments, for all DP models, the representations severely deteriorate, and the sensitivity space was orthogonal to that of non-DP models’ last layer (from where Grad-Cams are generated) (section 7.2; second last paragraph), as we discovered later.
>
> ``Assuming that the user/provider of the model can filter data points is very unrealistic in my opinion as only the DP model is available.``
>
> We do not have any user/provider filtering data points and it is a part of our experimentation. We first checked how the explanations, coming out of a DP model, are comparable with its non-DP counterpart, and till this point, we have both the models and the test samples apriori for the analysis. The relevance of a user/provider is exclusively coming in the software pipeline (section 9) when we have already gathered our findings from our extensive experimentations.

---

> ### Author Response · Authors · 2025-04-05
> **Authors's Response (Part 2/n)**
>
> W6+RC2
>
> ``In my opinion it doesn't make sense to use the non-private model for predicting and using it for the explanations as the model is then still susceptible to MIAs. This should be clearly discussed and made clearer in my opinion.``
>
>
> ``The main question of the paper is whether private model explanations can be given using DP models. However, this question is not entirely answered in my opinion. Instead, it is proposed to keep using the non-DP model and modify the explanation to be private. However, this defeats the purpose of using DP models in the first place, when trying to protect against MIAs and privacy attacks. As a result, I think either the research question needs to be adjusted to something like "Can DP models be used to generate private gradient-based explanations?" or another solution has to be proposed, being able to only use the DP model.``
>
>
> We carefully note your point. We have answered our research question precisely. However, to enhance the coherence, we have made a few modifications in our writing in section 6’s and 7.2’s last paragraph.
>
> We have previously clarified that reconciling explainability and privacy is not possible with DP models, and therefore, we looked for an alternative using LDP to make useful private explanations. However, as we cannot use the DP models and fall back to non-DP model, we clearly have to make some compromises, and there are a few silent factors to take care of when we have to reconcile. In the revised version, we have thoroughly rewritten section 9’s ‘Core processing pipeline’ and clearly explained the compromises for reconciliation in section 9’s last paragraph. Furthermore, we want to clarify that our proposition is solely based on our findings and setup, which we also clarified in section 9 (footnote 14).
>
> W3:
>
> ``The hypothesis that explainers perform poorly on DP models is only tested on 2 datasets (X-Ray and CIFAR-10)``
>
> We have trained 21 models (6 epsilon values x 3 types of model + 3 non-pvt models) for our main case study. Additionally, we have trained 9 extra models for CIFAR (3 epsilon values x 2 types of models + 2 non-pvt models (and also non-pvt efficientNet but DP models we couldn’t train)), all under a closely controlled environment and obtained similar conclusions on PIS across models. Furthermore, we have also investigated their representations and sensitivity space layerwise and conducted statistical significance test on representations, computed the similarity b/w pvt and non-pvt models and lastly conducted another round of significance test on the gradient space for all layers across privacy budgets (ϵ). If your acceptance still hurts due to # of datasets, we will run experiments on another dataset, if prescribed.
>
>
> RC3:
>
> ``As far as I can understand, no formal guarantees for the privacy of the LDP-explanation can be given. This should be clearly mentioned and discussed.``
>
> We LDP-fy with Laplacian noise, which, by definition, is ϵ-DP (section 8). We only had to fix the sensitivity and ϵ to make it utility preserving. However, in the previous version, we had a typo: we put an extra epsilon in the expression of sensitivity when dealing with pictures having c channels: (should be (k-1)nc/b^2, we wrote (k-1)nc/b^2ϵ; n = 1 in our case). We have fixed this in the new version, and with that, according to the definition of LDP, it is ϵ-DP. This notion can be parameterized by only ‘n’ or the number of maximum differing pixels, but Fan et al parameterized it with (n, b) to offer flexibility while preserving the utility while LDP-fying for a given ϵ. For more clarification beyond this point, we refer to Theorem 1 by Fan et al [1].
>
> RC5:
>
> ``Please go through the paper once more and fix all the typos and improve the transitions of the sentences.``
>
> done!
>
> [1] Liyue Fan. Image pixelization with differential privacy. In Data and Applications Security and Privacy XXXII: 32nd Annual IFIP WG 11.3 Conference, DBSec 2018, Bergamo, Italy, July 16–18, 2018, Proceedings 32, pages 148–162. Springer, 2018.

---

### Review · Reviewer_s3Qw · 2025-03-27

**Summary Of Contributions:**

This paper systematically investigates the relationship between privacy and explainability by analyzing the differences in explanation outputs between original models and their differentially private counterparts. It offers empirical insights into how differential privacy mechanisms influence model explainability, shedding light on the interplay between these two important aspects in high-stakes machine learning applications.

**Audience:**

Yes

**Broader Impact Concerns:**

No concerns.

**Claims And Evidence:**

No

**Requested Changes:**

- Include a more detailed survey and comparison of existing works that investigate the intersection of privacy and explainability, particularly those using differential privacy.

- Provide a comparative analysis of the proposed faithfulness evaluation with established methods such as ROAR and ROAD.

- Extend the experimental evaluation to include black-box explanation methods such as LIME and SHAP.

**Strengths And Weaknesses:**

**Strengthens:**

- The paper addresses an important and timely problem. As concerns around both explainability and privacy continue to grow in the machine learning community, it is valuable to investigate how these two dimensions interact.

- The paper is clearly written, and logically structured. I enjoy reading the paper.

**Weaknesses:**

- The paper claims in the introduction that the intersection of privacy and explainability has been largely unexplored. Although several related works are briefly mentioned in the related work section, a more thorough discussion is recommended. In particular, Patel et al. explicitly investigate the impact of differential privacy on explainability, which closely aligns with the focus of this paper. Providing a more comprehensive survey and a detailed comparison with these existing studies would better contextualize the paper’s contributions and clarify its novelty.

- While the paper critiques existing explanation faithfulness metrics and introduces its own, it overlooks several representative approaches such as ROAR [r1] and ROAD [r2], which are widely used and demonstrate strong performance. A detailed comparison with these methods would help position the proposed evaluation more effectively and justify its use.

- The study focuses exclusively on white-box explanation methods, overlooking popular black-box approaches like LIME and SHAP. Although these methods have received criticism in several literature, they remain widely adopted across various real-world applications. Including these methods in the experimental analysis could broaden the applicability and practical relevance of the paper’s findings.

[r1] Hooker, Sara, et al. "A benchmark for interpretability methods in deep neural networks." Advances in neural information processing systems 32 (2019).

[r2] Rong, Yao, et al. "A Consistent and Efficient Evaluation Strategy for Attribution Methods." International Conference on Machine Learning. PMLR, 2022.

---

> ### Author Response · Authors · 2025-04-05
> **Authors's Response (Part 1/n)**
>
> We wholeheartedly appreciate your analysis of our paper. We’d like to address your concerns as follows:
>
> W1+RC1:
>
> ``The paper claims in the introduction that the intersection of privacy and explainability has been largely unexplored. ``
>
> Thank you for your comment, but we didn't mention the “intersection” of privacy and explainability throughout the introduction. Rather, we are investigating the “interplay” b/w privacy and explainability for high-stakes. To be more specific, we have acknowledged that “incorporation” of privacy and explainability, which we have clearly stated in our writing, has been underexplored. When we write “incorporation”, that doesn’t convey what is common (or maybe common challenges) in both privacy and explainability (as indicated by “intersection”), and we have also clearly mentioned that, “Although privacy and explainability aspects have been vastly explored independently, there is little to no work incorporating them together.” For a better understanding of what we try to convey, we have written that, “We explore the
> unique challenges that emerge when attempting to integrate privacy and explainability aspects in high-stakes applications. We investigate the underlying causes of these challenges, examine the trade-offs involved, and discuss key considerations necessary for developing frameworks to successfully incorporate these two critical aspects effectively.”
>
> ``Although several related works are briefly mentioned in the related work section, a more thorough discussion is recommended.``
>
> We understand that the related work section needed to be elaborated on more, which we have done now. The second paragraph briefs the PPXAI methods; the third paragraph discusses how our work is different from theirs. The fourth paragraph briefs the class of works that investigate “the quality of explanations in privacy-preserving environments”; the fifth paragraph discusses how our work is different from those.
>
> If one, for the sake of argument, takes the example of Saifullah et al. (103) and says, they also reported about the degradation of attribution score found in DP models, with a series of ad-hoc selected quality measures and explainers without any justification, our paper is not about any such degradation. We start by discussing the most prominent class of measures to measure “quality of explanations”:  “faithfulness”. Measuring faithfulness and evaluating the effectiveness of a faithfulness measure is hard and still an open problem due to the absence of ground truth reasoning for a model’s prediction. And the existing measures come with some other problems, which we detailed in Section 3.3. So, bypassing this route, we (1) propose a desideratum named LA, private explanations to follow strictly in a utility preserving setting; (2) discover that the off-the-shelf explainers do not respect LA; (3) mechanistically interpret the models to find the reasons (Section 7) behind it; (4) explore an alternate solution to make the explanations private, and finally (5) propose a software pipeline to reconcile privacy and explainability in high stakes. Our introduction section includes all these points.
>
> Regarding Patel et al.:  Patel et al. attempts to make explanations private which are specifically generated by perturbation-based black box methods (like LIME). These methods require separate datasets (generated by perturbing the given example, referred to as “explanation dataset” in the paper) to generate an explanation. They claim that their proposed optimization method makes the explanation (ϵ,δ)-DP subject to the “explanation dataset”, but provide (O(mϵ), δ)-differential privacy against the training data, where m is the size of the “explanation dataset” (which makes the bound exceptionally large). Although they experiment with 2-layer DP models to investigate the L1 distance given the explanations obtained by their PPXAI method from the DP and non-DP models with three privacy budgets in an ad-hoc manner, that does not make our work aligned with the focus of this work, given that their aim was to propose a perturbation based black box PPXAI method to protect the explanation dataset, as opposed to our aim explained in the previous paragraphs.

---

> ### Author Response · Authors · 2025-04-05
> **Authors's Response (Part 2/n)**
>
> W2+RC2:
>
> ``While the paper critiques existing explanation faithfulness metrics and introduces its own, it overlooks several representative approaches such as ROAR [r1] and ROAD [r2], which are widely used and demonstrate strong performance.``
>
> We did criticize the existing faithfulness measures, but we didn’t propose any faithfulness evaluation measure. Firstly, we promptly consolidated the flaws in the commonly used faithfulness metrics to strengthen our motivation, and then we proposed two remarks that clarify our stand. In remark 2, we have stated that:
> “Explanations should align with local constraints and contexts, even when (so-called) faithfulness cannot be measured reliably.”
>
> In our context, we aim to use a proxy model (DP model) in place of another model (non-DP model), so we’ve introduced LA (section 5). It is self-explanatory that LA and remark 2 are orthogonal to faithfulness, and we primarily aim to measure the similarity (not faithfulness) b/w the explanations coming from DP and non-DP models for inputs where prediction matches, under certain conditions. As PIS, with which we are quantifying LA, in essence, is a similarity metric, we cannot compare the same with any faithfulness metric (including ROAR, ROAD, etc). We note that the hypothesis of LA is inspired by a well-regarded postulate (the model assumption) in contemporary XAI literature, but LA, in itself, is not a faithfulness measure.
>
> However, we would like to emphasize that LA is pretty effective for the job it was designed for. It did show us that the explanations from DP models are substantially incomparable with their non-DP counterparts, which, in turn, has triggered our inquisition to investigate the internals of DP models, and we later figured out (in section 7) that: DP training was primarily meant to resist MIA w.r.t. the training data but the training goes much beyond the scope, and drastically alters the overall representation and sensitivity space of a model. Which, as demonstrated, makes the popular post-hoc methods nonfunctional.
>
> ROAD and ROAR are merely one or two more faithfulness measures. We provide enough relevant literature in this section, consolidating known shortcomings. Moreover, due to the absence of ground truth reasoning for a model’s prediction (the last paragraph of Section 3.3 has been elaborated on more to make this clearer to the community), the remark that they ``demonstrate strong performance`` should be taken with a grain of salt.
>
> W3+ RC3:
>
> ``The study focuses exclusively on white-box explanation methods, overlooking popular black-box approaches like LIME and SHAP.``
>
> We didn’t “overlook” black-box explainers. We acknowledge our apprehension regarding the suitability of these explainers with black-box explanations, and they are not widely adopted in computer vision compared to the ones we have taken into consideration (section 5.3). We have noted the 3rd point of the requested changes, and for more clarity on the topic, we have rewritten the second paragraph of the same.
>
> Next, in our work, we also mechanistically interpret the interplay between the explainers and the DP model (Section 7). Even for the sake of discussion, if we include those explainers, we wouldn’t have been able to explain their behavior in our study. All in all, neither do we know whether we can use those explainers for our analysis, and even if we did use them, we couldn’t interpret their behavior. Therefore, we argue that unnecessarily trying to add these explainers wouldn’t make our study more scientific, and thus, we note this as a ‘feature’, not a bug in our study.

---

### Decision · Action_Editor_Ymkx · 2025-05-19

**Recommendation:** Accept as is

**Comment:**

All reviewers are happy with the authors' responses to their reviews and recommend acceptance.

**Audience:**

All reviewers agree that the submission would be interesting to the privacy and XAI communities.

**Claims And Evidence:**

All reviewers agree that the submission is supported by sufficient evidence.